# UHPLC-Orbitrap-MS Tentative Identification of 51 Oleraceins (Cyclo-Dopa Amides) in *Portulaca oleracea* L. Cluster Analysis and MS^2^ Filtering by Mass Difference

**DOI:** 10.3390/plants10091921

**Published:** 2021-09-15

**Authors:** Yulian Voynikov, Paraskev Nedialkov, Reneta Gevrenova, Dimitrina Zheleva-Dimitrova, Vessela Balabanova, Ivan Dimitrov

**Affiliations:** 1Department of Chemistry, Faculty of Pharmacy, Medical University of Sofia, 2 Dunav Str., 1000 Sofia, Bulgaria; idimitrov@pharmfac.mu-sofia.bg; 2Department of Pharmacognosy, Faculty of Pharmacy, Medical University of Sofia, 2 Dunav Str., 1000 Sofia, Bulgaria; pnedialkov@pharmfac.mu-sofia.bg (P.N.); rgevrenova@pharmfac.mu-sofia.bg (R.G.); dzheleva@pharmfac.mu-sofia.bg (D.Z.-D.); vbalabanova@pharmfac.mu-sofia.bg (V.B.)

**Keywords:** orbitrap, purslane, oleracein, diagnostic ion, diagnostic difference, clustering

## Abstract

Oleraceins are a class of indoline amide glycosides found in *Portulaca oleracea* L. (Portulacaceae), or purslane. These compounds are characterized by 5,6-dihydroxyindoline-2-carboxylic acid *N*-acylated with cinnamic acid derivatives, and many are glucosylated. Herein, hydromethanolic extracts of the aerial parts of purslane were subjected to UHPLC-Orbitrap-MS analysis, in negative ionization mode. Diagnostic ion filtering (DIF), followed by diagnostic difference filtering (DDF), were utilized to automatically filter out MS data and select plausible oleracein structures. After an in-depth MS^2^ analysis, a total of 51 oleracein compounds were tentatively identified. Of them, 26 had structures, matching one of the already known oleracein, and the other 25 were new, undescribed in the literature compounds, belonging to the oleracein class. Moreover, based on selected diagnostic fragment ions, clustering algorithms and visualizations were utilized. As we demonstrate, clustering methods provide valuable insights into the mass fragmentation elucidation of natural compounds in complex mixtures.

## 1. Introduction

*Portulaca oleracea* L. (Portulacaceae), or purslane, is a widely spread annual plant found in many parts of the world. Purslane is considered an edible vegetable in many areas of Europe, the Mediterranean, and tropical Asian countries, and is added in soups and salads [1,2,3]. Purslane has been used in folk and traditional medicine as a remedy for many ailments [4].

A number of reports exist in the literature on the ethnomedicinal and pharmacological profiles of purslane [4,5,6,7]. Purslane extracts demonstrate various pharmacological activities, such as antioxidant [2,8,9], anti-inflammatory [10,11,12], neuroprotective [13,14,15], antimicrobial [16,17], antidiabetic [18,19,20], antiulcerogenic [21], and anticancer [19,22,23].

Purslane contain diverse bio-active compounds, such as flavonoids, polysaccharides, fatty acids, vitamins and minerals, and alkaloids [1,2,4,5,24,25]. Among alkaloids, a class of indoline amide glycosides, called oleraceins, or cyclo-dopa amides, have been isolated in purslane extracts [12,25,26,27,28,29]. These compounds are characterized by 5,6-dihydroxyindoline-2-carboxylic acid *N*-acylated with cinnamic acid derivatives, and many are highly glucosylated (Table 1). All sugar moieties of identified oleraceins in the literature are β-D-glucopyranoses [12,27,29].

Xiang et al. [29] were the first to isolate and structurally characterize several of these indoline amides, naming them oleraceins A, B, C, and D. Later, Xing et al. [25] tentatively identified oleraceins A –D, F, and G, by Orbitrap-MS^2^ analysis in purslane extracts. Then, Liu et al. [30] isolated oleraceins F and G. Eight more oleraceins, H−O, were identified by Jiao et al. [26], based on UV and MS^2^ fragmentation analysis. A year later, the same team [27] isolated oleraceins H, I, K, L, N−S. Just recently, Fu et al. [12] isolated five new high molecular weight (>900 Da) oleraceins, naming them oleracein X, Y, Z, ZA, and ZB. In our previous study, 11 oleraceins (A–D, N–Q, S, U, and W glu) were tentatively identified [28].

Herein, utilizing UHPLC-Orbitrap-MS in negative ionization mode, we carried out an extensive tentative identification (level 2 annotation) of oleraceins in hydromethanolic extracts from the aerial parts of purslane. The MS^2^ characterization of oleraceins was limited to compounds having a mass up to 1 kDa, although heavier ones were also detected. After an in-depth MS^2^ analysis, a total of 51 oleracein compounds were tentatively identified and characterized, of which, 26 had structures, matching one of the already known oleraceins, and the other 25 were new, undescribed in the literature structures, belonging to the oleracein class. Diagnostic ion filtering (DIF), and diagnostic difference filtering (DDF), were utilized to refine the selection of compounds. Additionally, clustering of every oleracein based on their MS^2^ features was performed and presented with heatmaps, *k*-means and pam clustering, principal component analysis (PCA), and hierarchical clustering. As we demonstrate, clustering methods can provide valuable insights into the structure elucidation of natural compounds by mass spectrometry in complex mixtures.

## 2. Results and Discussion

A workflow diagram of the study is shown in Figure 1. In summary, hydromethanolic extract of purslane was obtained, and subjected to HR-MS^2^ analysis. The raw data were filtered by DIF, and then DDF, to select compounds possessing both specific fragment ions and specific mass differences corresponding to the presence of 5,6-dihydroxyindoline-2-carboxylic acid—a common scaffold for all oleraceins. The filtered MS^2^ data were then analyzed manually to structurally elucidate oleracein compounds. In the course of elucidation, 43 fragment ions were selected as diagnostic fragment ions that were afterwards used to describe every oleracein as a vector of length 43, and values equal to the relative percentage intensities of the diagnostic ions. This permitted to carry out clustering analyses to establish structural similarities between oleracein compounds, based on their MS^2^ features. The results from the clustering analysis were used to corroborate and supplement the structural elucidation or to correct it.

Oleraceins are characterized by 5,6-dihydroxyindoline-2-carboxylic acid *N*-acylated with either coumaric, caffeic, or ferulic acid. Most of them are 6-O glucosylated (Table 1). The sugar moieties of all characterized oleraceins in the literature are identified as *β*-D-glucopyranose [12,27,29], and so, in this paper, the hexose moieties are regarded as glucoses.

In our study, the lowest *m*/*z* oleracein detected was oleracein A, with a molecular ion of 502.135 [M–H]^−^
*m/z.* Our study limited the characterization of oleraceins with mass up to 1 kDa, although heavier oleraceins were also detected.

In total, 82 candidate substances were automatically selected, based on the above-mentioned criteria using DIF, followed by DDF, and their MS^2^ fragmentation manually inspected. The DDF results are presented in the Appendix A (Appendix A) as *m*/*z* transitions for all identified oleraceins. Of the total 82 candidates, 19 had too low MS^2^ intensity (base peak below 1.5×10^4^) and were not interpreted, 12 were false positives (not having oleracein structure), and 51 were identified as oleraceins (Table 2). Of them, 25 were new (undescribed in the literature) oleraceins. The other 26 compounds matched a structure of one of the already known oleraceins, namely: A, B, C, D (two isoforms), H, I (two isoforms), J, K, L, M (two isoforms), N/S (four isoforms), O (three isoforms), P, Q, W glu, X (three isoforms). Figure 2 presents the full-scan total ion chromatogram (TIC) and the extracted ion chromatogram (EIC) of the identified oleraceins. As observed, oleraceins are major components of purslane hydromethanolic extract. Table 2 presents the chemical structures of the identified 51 oleraceins and Table 3 provides their chromatographic and mass spectral characteristics. For complete chromatographic and mass spectral data, see Appendix A.

As mentioned earlier, all identified in the literature oleraceins are *N*-acylated with either coumaric, caffeic, or ferulic acid (i.e., bearing the substructure IC, IA, or IF). Each oleracein identified in our study is comprised of either GIC, GIA, or GIF alone, or linked with several other moieties that include hydroxybenzoyl (O), coumaroyl (C), caffeoyl (A), feruloyl (F), sinapoyl (S), and glucosyl (G). The presence of either of the following fragment ions indicate the type of hydroxycinnamic acid (HCA) *N*-linked to the indoline core: 340.0826 *m*/*z* (C_18_H_14_O_6_N^−^, [*N*-coumaroyl 5,6-dihydroxyindoline-2-carboxylic acid−H]^−^), 356.0765 *m*/*z* (C_18_H_14_O_7_N^−^, [*N*-caffeoyl 5,6-dihydroxyindoline-2-carboxylic acid−H]^−^), 370.0921 *m*/*z* (C_19_H_16_O_7_N^−^, [*N*-feruloyl 5,6-dihydroxyindoline-2-carboxylic acid−H]^−^) (Figure 3). In addition, the presence of each of the three HCAs can be confirmed by a set of fragments, including 145.0294 *m*/*z* for coumaroyl (C), 161.0244, and 109.0295 *m*/*z* for caffeoyl (A), and 175.0401, 161.02442, and 160.0166 *m*/*z* for feruloyl (F) moiety. The 5,6-dihydroxyindoline-2-carboxylic acid, or the indoline core, characteristic for every oleracein, is represented with fragment ions 194.0458, 150.0560, and 148.0403 *m/z,* in decreasing intensity. Fragment 194.0458 *m/z,* as well as the characteristic fragment ions for the HCAs are more prominent in oleraceins with lower mass. Regarding the fragmentation of the glycoside moieties, linked (consecutive) glucoses (as in oleraceins GGIC or GGGICG) are cleaved in the fragmentation process together. If consecutive neutral losses of hexoses (−162.053 Da, −C_6_H_10_O_5_) were observed (like in GICG), this suggested the presence of a single G substructure. In cases of two or three linked glucoses (such as in GGICGG or GGGICG), then neutral losses of 324.107 Da (−C_12_H_20_O_10_), or 486.159 Da (–C_18_H_30_O_15_), respectively, were observed.

### 2.1. Individual MS^2^ Fragmentation Analysis

Below, the tentative identification of all 51 oleraceins (Table 2 and Table 3) based on high-res MS^2^ fragmentation analysis is described, ordered by increasing mass. Characteristic neutral losses include: CO (−27.995 Da); CO_2_ (−43.990 Da); hydroxybenzoyl (O) (−120.020 Da); coumaroyl (C) (−146.037 Da); caffeoyl (A) (−162.032 Da); feruloyl (F) (−176.047 Da); sinapoyl (S) (−206.054 Da); glucosyl (G) (−162.053 Da); double glucosyl (GG) (−324.107 Da); triple glucosyl (GGG) (−486.159 Da).

The oleracein with the lowest *m*/*z* identified in our samples bears a precursor ion at 502.135 [M–H]^−^
*m*/*z* and elutes at rt 10.54 min. The MS^2^ spectrum displays a fragment ion 340.083 *m*/*z* (IC), formed after a G cleavage (−162.053 Da) from the molecular ion. This confirms the *N*-coumaroyl linkage (IC). The latter undergoes consecutive CO_2_ (−43.990 Da) and CO (−27.995 Da) cleavages, resulting in 296.093 and 268.098 *m/z,* respectively. The existence of the coumaroyl (C) is established by the fragment ions at 145.029, 119.049, and 117.033 *m/z.* The set of fragment ions at 194.046, 150.056, and 148.040 *m*/*z* confirm the presence of the indoline core (I). The proposed structure of this compound is glu-ind-coum, or GIC and concurs with the structure of oleracein A (Figure 4). 

The compound with precursor ion at 518.130 [M–H]^−^
*m*/*z*, identified as glu-ind-caff, or GIA, elutes at rt 9.07 min. Fragment ion at 356.078 *m*/*z* confirms the *N*-caffeoyl linkage (IA). Fragment 356.078 *m*/*z* (IA), unlike fragment 340.083 *m*/*z* (IC), reveals a cleavage of C_6_H_6_O_2_ (−110.037 Da) with consequent loss of CO_2_, transition 356.078 → 246.041 → 202.051 *m*/*z* (Figure 4). Fragment 109.028 *m*/*z* ([C_6_H_6_O_2_-H]^−^) confirms the previous finding. Fragment 356.078 *m*/*z* is capable of CO_2_ cleavage to form fragment 312.088 *m/z.* The A and I substructures are corroborated by their corresponding fragments. The structure of this compound matches oleracein W glu.

The MS^2^ data of the compound with 532.146 [M–H]^−^
*m*/*z*, eluting at rt 11.23, identified as glu-ind-fer, or GIF, reveals the set of fragment ions corroborating the IF structure, namely 370.093, 326.103, and 298.108 *m/z,* formed in a similar fashion as discussed for GIC, after consecutive CO_2_ and then CO losses, respectively, from IF. The structure of GIF coincides with oleracein B (Figure 4).

The base peak chromatogram of 664.188 [M–H]^−^
*m*/*z* reveals two peaks eluting at rt 7.32 and 9.00 min. The isobar eluting at 7.32 min is identified as glu-ind-coum-glu, or GICG. Fragment 340.083 *m*/*z* confirms the IC. The molecular ion can consecutively cleave the two proximal hexoses, resulting in the transition 664.188 → 502.135 → 340.083 (IC) *m/z.* Then, the fragmentation of 340.083 *m*/*z* is as described earlier for GIC. The proposed structure of GICG coincides with oleracein C. 

The other isobar eluting at rt 9.00 min is identified as glu-glu-ind-coum, or GGIC. Fragment ion 340.083 *m*/*z* confirms the IC. The molecular ion can cleave the C, resulting in fragment ion 518.151 *m*/*z* (GGI). The subsequent neutral loss of 324.107 Da suggests a joint GG loss, resulting in 194.046 *m*/*z* (I). The proposed structure of this compound matches oleracein H.

The compound with 678.183 [M–H]^−^
*m*/*z* eluting at rt 16.63 min is identified as fer-glu-ind-coum, or FGIC. Fragment ion 340.083 *m*/*z* confirms the IC. The molecular ion can cleave either the proximal C (−146.037 Da) or F (−176.047 Da), resulting in 532.146 *m*/*z* (FGI) or 502.135 *m*/*z* (GIC), respectively. In accordance with the proposed structure, fragments 532.146 and 502.135 *m*/*z* can undergo an I (−195.053 Da) or G (−162.053 Da) losses, resulting in 337.093 (FG) or 340.083 *m*/*z* (IC), respectively. 

The base peak chromatogram of 680.183 [M–H]^−^
*m*/*z* reveals two peaks eluting at rt 7.26 and 7.82 min. The compound eluting at rt 7.26 min is identified as glu-ind-caff-glu, or GIAG. Fragment 356.078 *m*/*z* (IA) undergoes the usual fragmentation as described earlier for GIA. The molecular ion initially cleaves either of the proximal G, transition 680.183 → 518.130 *m*/*z*. The latter is able to cleave another G, resulting in fragment ions 356.078 (IA) *m*/*z*. The isobar eluting at 7.82 min is identified as glu-glu-ind-caff, or GGIA. The molecular ion initially cleaves the A (−162.032 Da), transition 680.183 → 518.151 *m*/*z* (GGI). In contrary to its isobar GIAG, here, GGIA demonstrates a loss of GG (−324.107 Da) in the transition 680.183 → 356.078 *m*/*z* (IA). 

The base peak chromatogram of 694.199 [M–H]^−^
*m*/*z* reveals four isobars eluting at rt 5.91, 6.78, 8.05 and 9.64 min. The pair of compounds at rt 5.91 and 9.64 min had identical fragmentation behavior and are identified to have the structure of glu-glu-ind-fer, or GGIF. Fragment 370.093 *m*/*z* confirms the IF. The GG loss appears jointly in the transition 694.199 (GGIF) → 370.093 (IF) *m*/*z*. The proximal F is cleaved in the transition 694.199 → 518.151 (GGI). The proposed structure of the two isobars coincides with oleracein I. 

The other isobar pair eluting at 6.77 and 8.04 had identical fragmentation and are identified as having the structure glu-ind-fer-glu, or GIFG. Contrary to the GGIF, here, the neutral losses of each of the proximal G (2 × −162.053 Da) appear consecutively in the transitions 694.199 → 532.146 → 370.093 (IF) *m*/*z*. The fragmentation of 370.093 *m*/*z* is as described for GIF. The proposed structure of the two isobars coincides with oleracein D.

The base peak chromatogram of 708.193 [M–H]^−^
*m*/*z* reveals two isobars eluting at rt 16.41 and 16.89 min. The eluent at rt 16.41 min, is identified as sin-glu-ind-coum, or SGIC. Fragment 340.083 *m*/*z* confirms the IC. The molecular ion can cleave, consecutively, the proximal C, and then I, resulting in fragments 562.157 (SGI) and 367.103 *m*/*z* (SG), respectively. The latter can cleave a G, resulting in 205.051 *m*/*z* (S). The presence of a sinapoyl (S) is supported by the neutral loss of 206.054 Da in the transition 708.193 (SGIC) → 502.135 *m*/*z* (GIC), as well as the fragments 223.061 and 191.035 *m*/*z*. The fragmentation of 502.135 *m*/*z* proceeds as described for GIC.

The other isobar eluting at rt 16. 88 min is identified as fer-glu-ind-fer, or FGIF. Fragment 370.093 *m*/*z* confirms the IF. The molecular ion cleaves a F, transition 708.193 → 532.146 *m*/*z*, followed by either a G, or an I loss, resulting in fragments 370.093 (IF) or 337.093 (FG) *m*/*z*, respectively. 

The compound eluting at rt 15.05 min displays a molecular ion at 724.189 [M–H]^−^
*m*/*z* and is identified as sin-glu-ind-caff, or SGIA. Fragment 356.078 *m*/*z* confirms the IA. A neutral loss of the SG fragment appears (−368.110 Da) from the molecular ion, transition 724.189 → 356.078 *m*/*z*. The MS^2^ spectra in positive mode was also considered for analysis to further corroborate the proposed structure. The molecular ion at 726.203 [M+H]^+^
*m*/*z* cleaves the proximal A, followed by an I loss to produce fragments 564.173 (1.93) (SGI) and 369.117 (6.58) (SG) *m*/*z*, respectively. Fragment ion 358.090 (9.63) *m*/*z* confirms the IA. Fragment ion 369.117 *m*/*z* (SG) can cleave a G to produce 207.065 (100) *m*/*z* (S), which can fragment to 147.044 (0.68) *m*/*z*. A is also confirmed with fragment ion 163.038 (45.13) *m*/*z* and I is represented by 196.060 (10.85) *m*/*z*.

The compound with molecular ion at 738.204 [M–H]^−^
*m*/*z*, eluting at rt 16.66 min, is identified as sin-glu-ind-fer, or SGIF. The *N*-feruloyl moiety is confirmed by the fragment at *m*/*z* 370.093 (IF). The molecular ion displays initially either a F or S (−206.054 Da) loss, resulting in fragments 562.157 (SGI) or 532.146 (GIF) *m*/*z*, respectively, although the former is preferred. Fragment 532.146 *m*/*z* can cleave a G, resulting in 370.093 *m*/*z* (IF). Fragment ions 223.061 and 205.051 *m*/*z* corroborate the presence of S.

The compound with molecular ion at 784.188 [M–H]^−^
*m*/*z* eluting at rt 10.61 min is identified as hb-glu-glu-ind-coum, or OGGIC. The IC fragment is confirmed by the fragment 340.083 *m*/*z*. The molecular ion can cleave a proximal C, resulting in fragment 638.173 *m*/*z* (OGGI), which can undergo an I loss, resulting in fragment 443.119 *m*/*z* (OGG). The MS^2^ spectra revealed a major peak at 137.024 *m*/*z* that did not correspond to any previously described in the literature substructure common to oleraceins. Moreover, in the transition 638.173 → 518.151 *m*/*z* there is a loss of 120.020 Da that suggested a dehydrated derivative of fragment 137.024 *m*/*z*. This led us to propose that fragment 137.024 *m*/*z* corresponds to a hydroxybenzoyl moiety (O). Fragment 137.024 can cleave a CO_2_ to result in fragment 93.033 *m*/*z* (not shown).

The compound with molecular ion at 810.225 [M–H]^−^
*m*/*z* eluting at rt 11.78 min is identified as coum-glu-glu-ind-coum, or CGGIC. The molecular ion displays initial cleavage of proximal C, transition 810.225 → 664.190 *m*/*z* (GGIC). The further fragmentation is identical to that of GGIC.

The compound with molecular ion at 826.220 [M–H]^−^
*m*/*z* eluting at rt 10.59 min is identified as caff-glu-glu-ind-coum, or AGGIC. Fragment 340.083 *m*/*z* confirms the IC. The molecular ion can cleave either a C or an A, resulting in fragment ions 680.183 *m*/*z* (AGGI), 664.188 *m*/*z* (GGIC) and 518.151 *m*/*z* (GGI). Fragment 485.130 *m*/*z* (AGG) is also observed. The proposed structure coincides with oleracein K.

The compound with molecular ion at 826.241 [M–H]^−^
*m*/*z* eluting at rt 6.34 min is proposed to have the structure of glu-glu-ind-coum-glu, or GGICG. The molecular ion initially cleaves the single G after which the resulting fragment at 664.188 *m*/*z* (GGIC) can cleave a C, resulting in 518.151 *m*/*z* (GGI). Fragment 502.135 *m*/*z* (ICG) is formed after a cleavage of the GG from the molecular ion. Fragment 518.151 *m*/*z* also displays GG loss, resulting in 194.046 *m*/*z* (I). The proposed structure coincides with oleracein P.

The base peak chromatogram of 840.236 [M–H]^−^
*m*/*z* reveals five isobars eluting at rt 10.17, 11.95, 12.36, 13.31, and 14.09 min. Except the eluent at rt 13.35 min, all other isobars demonstrate identical fragmentation and are identified as fer-glu-glu-ind-coum, or FGGIC, which coincide with the structure of oleracein N or oleracein S. Fragment ion 340.083 *m*/*z* confirms the IC. The molecular ion is able to cleave either the proximal C or F, resulting in the transition 840.236 → (694.199 or 664.188) → 518.151 *m*/*z* (GGI). The latter can cleave the GG, resulting in 194.046 *m*/*z* (I). Fragment 694.199 (FGGI) is able to cleave an I, resulting in fragment ion 499.146 *m*/*z* (FGG). The proposed structure for the isobar eluting at 13.32 is fer-glu-ind-coum-glu, or FGICG. Initially, the molecular ion cleaves the proximal G to produce 678.183 *m*/*z* (FGIC), which can either cleave a C or F, to produce 532.146 (FGI) and 502.135 *m*/*z* (GIC), respectively. The FGI fragment ion at 532.146 *m*/*z* can in turn lose an I, resulting in 337.093 *m*/*z* (FG), whereas fragment 502.135 *m*/*z* (GIC) can lose a C, resulting in 340.083 *m*/*z* (IC).

The compound with molecular ion at 842.215 [M–H]^−^
*m*/*z*, eluting at rt 9.34 min, is proposed to have the structure of caff-glu-glu-ind-caff, or AGGIA. The molecular ion initially cleaves either of its two proximal caffeoyls, to give 680.183 *m*/*z*. If the *N*-caffeoyl is cleaved, then the I can be cleaved (−195.053 Da), resulting in fragment ion 485.130 *m*/*z* (AGG). Fragment 518.151 *m*/*z* (GGI) occurs when both proximal A are cleaved. In the other fragmentation pathway, where the initially cleaved caffeoyl is the one linked to GG, the stripped hexoses can be cleaved in concert from fragment 680.183 *m*/*z*, to give fragment 356.078 *m*/*z* (IA).

The compound with molecular ion at 842.236 [M–H]^−^
*m*/*z*, eluting at rt 6.37 min is proposed to have the structure of glu-glu-ind-caff-glu, or GGIAG. Fragment 356.078 *m*/*z* confirms the IA. Fragment ion 680.183 *m*/*z* (GGIA) is produced by the cleavage of the proximal G from the molecular ion. Fragment ion 518.145 *m*/*z* is actually composed of two closely *m*/*z* spaced fragment ions: 518.130 *m*/*z* (GIA) and 518.151 *m*/*z* (GGI). The overlapping of these fragment ions results in the appearance of the 518.145 *m*/*z*. Fragment ion 518.130 *m*/*z* results from GG cleavage (−324.107 Da) from the molecular ion, whereas 518.151 *m*/*z* occurs after consecutive cleavage of the single proximal G followed by the A cleavage (−324.085 Da total).

The base peak chromatogram of 856.231 [M–H]^−^
*m*/*z* reveals two isobars eluting at rt 10.68 and 11.08 min. The compound eluting at rt 10.68 min is identified as fer-glu-glu-ind-caff, or FGGIA. Fragment 356.078 *m*/*z* confirms the IA. The molecular ion displays a cleavage of proximal A, resulting in fragments 694.199 *m*/*z* (FGGI). The latter can cleave the proximal F to result in 518.151 *m*/*z* (GGI), which in turn, after cleavage of GG, results in I (194.046 *m*/*z*). The proposed structure coincides with oleracein J. The isobar eluting at rt 11.08 min is identified as caff-glu-glu-ind-fer, or AGGIF. Fragment 370.093 *m*/*z* confirms the IF. The molecular ion displays consecutive cleavages of F and A, irrespective of the order, resulting from the transitions 856.231 → (694.199 or 680.183) → 518.151 *m*/*z* (GGI). Fragment 680.183 *m*/*z* (AGGI) can cleave an I, resulting in 485.130 *m*/*z* (AGG). Moreover, the latter can cleave GG, resulting in fragment at 161.024 *m*/*z* (A). The proposed structure coincides with oleracein L.

The base peak chromatogram of 856.252 [M–H]^−^
*m*/*z*, eluting at rt 7.04 min is identified as glu-glu-ind-fer-glu, or GGIFG. Fragment at 370.093 *m*/*z* confirms the IF. The molecular ion can cleave the single G to result in fragment 694.199 *m*/*z* (GGIF) that, in turn, can cleave a F to result in fragment 518.151 *m*/*z* (GGI). Fragment 532.146 *m*/*z* (GIF) is formed after the GG loss from the molecular ion. The proposed structure coincides with oleracein Q.

The base peak chromatogram of 870.246 [M–H]^−^
*m*/*z* reveals eight isobars eluting at rt 10.65, 11.71, 12.40, 12.79, 13.20, 13.52, 13.98, and 14.33 min. The isobars eluting at rt 10.65, 12.40 and 12.79 min have identical MS^2^ fragmentation and are identified as fer-glu-glu-ind-fer, or FGGIF. Fragment at 370.093 *m*/*z* confirms the IF. The molecular ion cleaves a proximal F resulting fragment 694.199 *m*/*z* which can either cleave the other F, or an I, resulting in fragments 518.151 (GGI) or 499.146 *m*/*z* (FGG). Both these fragments possess proximal GG that can be cleaved, resulting in fragments 194.046 (I) and 175.040 *m*/*z* (F), respectively. The proposed structure coincides with oleracein O. The isobars eluting at rt 13.52 and 14.33 min have identical MS^2^ fragmentation and are identified as glu-fer-glu-ind-fer, or GFGIF. Fragment at 370.093 *m*/*z* confirms the IF. Initially, the molecular ion cleaves a G, resulting in fragment 708.193 *m*/*z* (FGIF). The latter can cleave either of the proximal F giving fragment 532.146 *m*/*z*. Depending on the feruloyl cleaved, the latter fragment can either cleave an I or a G, resulting in fragments 337.093 (FG) or 370.093 *m*/*z* (IF), respectively. The isobars eluting at rt 11.71 and 13.98 min have identical MS^2^ fragmentation and are identified as sin-glu-glu-ind-coum, or SGGIC. Fragment 340.083 *m*/*z* confirms the IC. The molecular ion is able to cleave either C or S, resulting in the transitions 870.246 → (724.209 (SGGI) or 664.188 (GGIC)) → 518.151 *m*/*z* (GGI). The latter fragment can cleave the GG, resulting in fragment 194.046 *m*/*z* (I). The proposed structure matches oleracein M. The isobar eluting at rt 13.20 min is identified as sin-glu-ind-coum-glu, or SGICG. Fragment 340.083 *m*/*z* confirms the IC. The molecular ion exhibits a proximal G cleavage, resulting in fragment 708.193 *m*/*z* (SGIC), which in turn can cleave either a C or a S, resulting in fragments 562.157 (SGI) or 502.135 *m*/*z* (GIC), respectively. Fragment 562.157 *m*/*z* (SGI) can cleave the stripped I, resulting in 367.103 *m*/*z* (SG).

The compound eluting at rt 10.43 min displays a molecular ion at 886.241 [M–H]^−^
*m*/*z* and is identified as sin-glu-glu-ind-caff, or SGGIA. Fragment 356.078 *m*/*z* confirms the IA. The molecular ion exhibits a proximal A cleavage, resulting in fragment 724.209 *m*/*z* (SGGI). The latter can either cleave an I or a S, resulting in fragments 529.156 (SGG) or 518.151 *m*/*z* (GGI), respectively. Both these fragments can lose the proximal GG in concert, resulting in a S (202.051 *m*/*z*) or an I (194.046 *m*/*z*) fragment ions.

The base peak chromatogram of 900.256 [M–H]^−^
*m*/*z* reveals four isobars eluting at rt 10.56, 12.12, 13.41, and 14.09 min. All compounds but the eluent at rt 13.41 min show identical fragmentation and are identified as sin-glu-glu-ind-fer, or SGGIF. The molecular ion is able to cleave either proximal F or S, resulting in the transitions 900.256 → (724.209 (SGGI) or 694.199 (GGIF)) → 518.151 *m*/*z* (GGI). Fragment 724.209 *m*/*z* can lose the stripped I, resulting in fragment at 529.156 *m*/*z* (SGG). Fragment 529.156 and 518.151 *m*/*z* can both cleave their GG, resulting in fragments 205.051 (S) or 194.046 *m*/*z* (I). The proposed structure matches oleracein X.

The isobar eluting at rt 13.41 min is identified as glu-sin-glu-ind-fer, or GSGIF. Fragment at 370.093 *m*/*z* confirms the IF. After the proximal G cleavage from the molecular ion, the resultant fragment at 738.205 *m*/*z* (SGIF) is able to cleave either F or S, resulting in fragments 562.157 (SGI) and 532.146 *m*/*z* (GIF), respectively. Fragment 562.157 *m*/*z* can lose the I, resulting in fragment 367.103 *m*/*z* (SG) which in turn can lose a G to give the S fragment featured at 205.051 *m*/*z*. Fragment 532.146 *m*/*z* can lose the proximal G to give fragment 370.093 *m*/*z* (IF).

The compound eluting at rt 8.65 min having a molecular ion at 946.262 [M–H]^−^
*m*/*z* is identified as hb-glu-glu-ind-coum-glu, or OGGICG. Fragment 340.083 *m*/*z* confirms the IC. The molecular ion initially cleaves the single G to give fragment 784.215 *m*/*z* (OGGIC) which in turn can lose a C to give fragment 638.173 *m*/*z* (OGGI). The latter can lose an I to result in 443.119 *m*/*z* (OGG). Fragment 638.173 *m*/*z* can also lose a hydroxybenzoyl moiety (O) (−120.020 Da) to result in 518.151 *m*/*z* (GGI). The latter can cleave GG, to result in 194.046 *m*/*z* (I). Similar to oleracein OGGIC described before, here we observe fragment ion 137.024 *m*/*z* (O). The decarboxylated fragment of 137.024 *m*/*z* is also observed at 93.033 *m*/*z* (not shown). The MS^2^ data in positive ionization mode shows a dehydrated hydroxybenzoyl derived fragment ion at 121.029 *m/z.*

The base peak chromatogram of 988.273 [M–H]^−^
*m*/*z* reveals four isobars eluting at rt 7.78, 8.52, 9.45, and 10.07 min. The compound eluting at 7.78 min is identified as caff-glu-glu-ind-coum-glu, or AGGICG. Fragment at 340.083 *m*/*z* confirms the IC. Initially, the molecular ion is observed to make the transition 988.273 → 826.228 *m*/*z* (−162.045 Da), where the difference of 162.045 Da is a weighted average of a G (−162.053 Da) and A (−162.032 Da) loss. Accordingly, the abovementioned fragment ion 826. 228 *m*/*z* is a weighted average of 826.220 (AGGIC) and 826.241 *m*/*z* (GGICG). Afterwards, 664.188 *m*/*z* (GGIC) fragment ion is observed. Fragment 680.183 *m*/*z* (AGGI) results from consecutive G and C losses from the molecular ion and is able to cleave the stripped I to produce 485.130 *m*/*z* (AGG). Fragment 664.188 *m*/*z* as well as 485.130 *m*/*z* both possess proximal GG that can be cleaved in concert, resulting in 340.083 (IC) and 161.024 *m*/*z* (A), respectively. The isobars eluting at rt 8.52 and 9.45 min demonstrate identical fragmentation behavior and are identified as glu-caff-glu-glu-ind-coum, or GAGGIC. Fragment ion 340.083 *m*/*z* confirms the IC. The molecular ion can lose a C and G, resulting in the transitions 988.273 → (842.236 or 826.220) → 680.183 *m*/*z* (AGGI). Fragment ion 826.220 *m*/*z* (AGGIC) can lose the A to result in 664.188 *m*/*z* (GGIC). Fragment ion 680.183 *m*/*z* can cleave A or I, resulting in 518.151 (GGI) or 485.130 *m*/*z* (AGG), respectively. The latter two fragments can both cleave GG to result in fragment ions 194.046 (I) and 161.024 (A) *m*/*z*, respectively. The isobar eluting at rt 10.07 min is identified as caff-glu-glu-glu-ind-coum, or AGGGIC. Fragment ion 340.083 *m*/*z* confirms the IC. The molecular ion can lose the proximal C or A, resulting in the transitions 988.273 → (842.236 or 826.241) → 680.204 *m*/*z* (GGGI). Here, the three hexoses are conjoint, and thus, their cleavage is observed in concert (−486.159 Da) from fragment 680.204 *m*/*z*, resulting in 194.046 *m*/*z* (I). Accordingly, fragment 826.241 *m*/*z* (GGGIC) also demonstrates the potential of the GGG loss, transition 826.241 → 340.083 *m*/*z* (IC).

The base peak chromatogram of 988.294 [M–H]^−^
*m*/*z* reveals three isobars eluting at rt 5.83, 6.03 and 6.21 min. The compound eluting at rt 5.83 min is identified as glu-glu-ind-coum-glu-glu, GGICGG. Fragment 340.083 *m*/*z* confirms the IC. In accordance with the proposed structure, the molecular ion can cleave GG on either side of the molecule, resulting in the transitions 988.294 → 664.188 → 340.083 *m*/*z* (IC). If the cleaved GG are linked to the coumaroyl, the C is observed to be cleaved, resulting in the transition 664.188 → 518.151 *m*/*z* (GGI). The latter can cleave GG to result in fragment 194.046 *m*/*z* (I). The isobars, eluting at rt 6.03 and 6.22 min, display identical fragmentation and are identified as glu-glu- glu-ind-coum-glu, or GGGICG. Fragment ion 340.083 *m*/*z* confirms the IC. The transition 988.294 → 826.241 (GGGIC) → 680.204 *m*/*z* (GGGI) corresponds to consecutive losses of the proximal single G followed by a C loss from the molecular ion. The loss of the GGG is observed in the transition 988.294 → 502.135 *m*/*z* (ICG) as well in the transition 680.204 → 194.046 *m*/*z* (I). Then fragment 502.135 *m*/*z* (ICG) can lose the other G to result in 340.083 *m*/*z* (IC).

### 2.2. Diagnostic Ions

Since oleracein compounds bear similar structure, we sought to refine the “fragment ion pool” and to select diagnostic fragment ions that can be used to describe the identified oleraceins. Thus, after thorough MS^2^ fragmentation analysis, we selected 43 fragment ions, their elemental compositions and exact masses determined, that were utilized as diagnostic ions for the identified substances (Appendix A). Hence, each oleracein was described as a vector of length 43 with values equal to the relative intensities of the corresponding diagnostic ions. A fragment ion from an MS^2^ data of a particular oleracein was assigned to a diagnostic ion if its *m*/*z* were within 15 ppm error of the diagnostic ion’s *m/z.* All diagnostic ions had the following features: a mass greater than 100 Da, were encountered in two or more oleraceins, had a mean percent intensity greater than 5%, and a maximum percent intensity greater than 10%. The diagnostic fragment ions along with their featured structures are shown in Appendix A and discussed below in increasing mass.

Fragment ion 137.0243 *m*/*z* is derived from the hydroxybenzoyl (O) moiety, as described in the characterization of OGGIC and OGGICG. The coumaroyl (C) moiety can be evident by fragment ion 145.0294 *m*/*z*. The caffeoyl (A) can be confirmed with fragment ion 161.0243 *m/z;* however, if not linked to the indoline core (I), fragment 179.0349 *m*/*z* may as well be present. Fragment 161.0243 *m*/*z* might indicate a feruloyl (F) as well; however, in our study, the caffeoyl produced > 60%, whereas the feruloyl produced < 30% intensity of fragment ion 161.0243 *m*/*z*. Fragment ion 175.0400 *m*/*z* indicates the presence of F. The appearance of fragment 193.0506 *m*/*z* might indicate F not linked to I. The I is featured by fragment ions 194.0458, 150.0560 and 148.0403 *m*/*z*, in decreasing intensity. Fragment 194.0458 *m*/*z* as well as the characteristic fragments for the HCAs are usually very prominent, their intensity decreases with increasing the mass of the molecule, unless the molecule possesses easily cleavable moieties, like consecutive glucoses, as in GGGICG or GGICGG. Fragment 205.0505 *m*/*z* is observed in all identified substances, bearing the sinapoyl (S) moiety, as well as fragment 223.0611 *m*/*z*, in lower intensity. As the S is encountered only linked to a glucosyl (G), fragment ions 562.1565 and 367.1034 *m*/*z* corresponding to the SGI and SG substructures, respectively, can indicate the presence of S. 

As mentioned above, the ind-HCA structures: IC, IA, and IF, are confirmed with their corresponding fragment ions: 340.0826, 356.0775, and 370.0931 *m*/*z*, respectively. However, as the mass of the oleracein increases, the intensities of these characteristic fragments might lower. If the oleraceins bear easily cleavable moieties, like two or three consecutively linked G, the fragment ions indicating the ind-HCA structures may be more prominent. Thus, in the oleracein GGGICG, where the GGG cleave together as a neutral loss of 486.159 Da, high intensity of fragment 340.0826 *m*/*z* (64%) is observed, as well as fragment 145.0294 *m*/*z* (100%). On the other hand, in the oleracein AGGIC, lower intensities of both these fragments are observed (11% and 37%, respectively).

The IC fragment at 340.0826 *m*/*z* undergoes consecutive CO_2_ and CO cleavages, resulting in fragments 296.0927 and 268.0978 *m*/*z*, respectively, in decreasing intensity. The same fragmentation behavior is established for the IF fragment at 370.0931 *m*/*z*, where the transition 370.0931 → 326.1033 → 298.1084 *m*/*z* is observed. The fragmentation of IA at 356.0775 *m*/*z*, however, is somewhat different. Very little if any of the CO_2_ and then CO losses could be observed from the IA fragment. Instead, 356.0775 *m*/*z* loses C_6_H_6_O_2_ (−110.036 Da), which corresponds to the dihydroxybenzene moiety from the caffeoyl. Additionally, the dihydroxybenzene fragment could be confirmed at 109.0295 *m*/*z*. The resultant fragment at 246.0407 *m*/*z* then loses CO_2_ to produce fragment 202.0509 *m*/*z*. Thus, in IA, the transition 356.0775 → 246.0407 → 202.0509 *m*/*z* is established, and the intensities are higher than the abovementioned daughter fragment ions for IC and IF. The rest of the diagnostic ions and their features are listed in Appendix A.

### 2.3. Clustering Methods

The clustering methods express the similarity between the oleraceins based on their MS^2^ features. Initially, a *m* by *n* ions matrix was created, describing every oleracein (rows, *m* = 51) with their corresponding diagnostic ions (columns, *n* = 43), shown in Appendix A. Every cell in the matrix represents the percentage intensity of a diagnostic ion for a particular oleracein. If a diagnostic ion was missing in the MS^2^ data of a compound, zero intensity was assigned. This data were imported to RStudio and manipulated further with the R programming language. Different methods were used to cluster the oleraceins based on their MS^2^ features. As a primary step, a distance matrix was created (*51 × 51*), calculating the Euclidean distance from the ions matrix. Figure 5 represents the ordered and unordered heatmaps, using the data from the distance matrix.

Then, *k*-means and pam clustering were used to estimate the optimal number of clusters. The clustering observed in the ordered heatmap (Figure 5), as well as the data of the *k*-means and pam clustering (Appendix A), suggested to cluster our data with eight clusters (Figure 6). Distribution of individuals in the groups can be found in Appendix A. 

Next, principal components were calculated. Scree plot (representing the percentage of variances explained by each principal component) can be found in Appendix A. The PCA visualization is presented in Figure 7, where the color gradient from orange (darker) to blue (lighter) presents the quality of representation (cos2), from high to low. A high cos2 indicates a good representation of the variable on the principal component, and vice versa. Hence, three groups can be distinguished, that are characterized by either IA (1st quadrant), IF (2nd quadrant) or IC (4th quadrant) substructures. The quality of representation (cos2) of individuals as well as a visualization of the contribution of individuals on PC1 and PC2 are given in Appendix A.

Figure 8 depicts the clustering of the individual fragment ions, positively correlated variables (fragment ions) point to the same side of the plot, and vice versa. Thus, we can clearly observe which fragment ions “go together” (are parent ion → daughter ion). As expected, identical to Figure 7, the fragment ions corresponding to IA (1st quadrant), IF (2nd quadrant), and IC (4th quadrant), are observed as they are described in the “diagnostic ions” section, and presented in Figure 3.

Next, hierarchical clustering was employed with method = “average”. The cophenetic correlation coefficient was calculated to be 0.85. The hierarchical clustering is visualized as a dendrogram and a phylogenetic tree in Figure 9, where the tree was “cut” into 8 parts.

Overall, the different clustering methods used resulted in similar clustering. In accordance with the proposed structures, different isoforms with the same structure are clustered together (as FGGIC, FGGIC.1, FGGIC.2, etc.). In general, oleraceins are grouped depending on the presence of either of the three common substructures: GIC, GIA, or GIF, that give rise to other diagnostic fragment ions. Thenceforward, different combinations or permutations of substructures lead to specific diagnostic fragment ions.

Oleraceins GIC, FGIC, SGIC, GICG, FGICG, and SGICG, have either GIC or GICG in common, and are grouped together. Next, GGICGG has the unique feature that there is a GG attached to the *N*-coumaroyl. It does, however, show some similarity to GGICG, GGGICG, GGGICG.1. The latter three oleraceins cluster together, with the ICG substructure in common. Next, a cluster composed of GGIC, CGGIC, FGGIC, FGGIC.1, FGGIC.2, FGGIC.3, SGGIC, and SGGIC.1, is formed with the GGIC as a common feature, and no G attached to the *N*-coumaroyl. The two representatives possessing O are clustered together; fragment ion 137.0243 *m*/*z* corresponding to the O substructure is exhibited at high intensity and unique to OGGIC and OGGICG. Oleracein AGGICG shows structural uniqueness, but nevertheless, demonstrates similarity with AGGIC, GAGGIC, and GAGGIC.1, which conforms with their proposed structures. Another cluster comprised of AGGIA, AGGIF, AGGIC, GAGGIC and GAGGIC.1 is formed with the AGGI substructure in common. The A not attached to I is indicated by the appearance of fragment ion 179.035 *m*/*z*, in addition to 161.0243 *m*/*z*, characteristic for A. Moreover, fragment ion 680.1831 *m*/*z*, that indicates GGIA or GIAG or AGGI, is exhibited in prominent intensity. AGGGIC shows distinctive MS^2^ features due to, on one hand, the proximal A, and on the other hand, the GGG. Other clustered oleraceins bearing the GIA substructure include: GGIA, FGGIA, and SGGIA with the GGIA substructure in common. GIAG and GGIAG, as well as GIA and SGIA, are clustered pairs, which is also evident from their structures. Oleraceins FGGIF, FGGIF.1, FGGIF.2, GGIF, GGIF.1, SGGIF, SGGIF.1, and SGGIF.2 are grouped with GGIF as a common substructure. Another grouping is formed between oleraceins with the GIFG common structural feature, namely GIFG, GGIFG, and GIFG.1. The rest of the oleraceins bearing the GIF do not possess G attached to the *N*-feruloyl, have a single G attached to I, and not GG or GGG. They are grouped and represented with oleraceins GIF, FGIF, GFGIF, and SGIF.

It is worth mentioning that the calculated similarities do not provide direct quantitative measure of the structural similarity. Some substructures (or moieties) are represented with more than one diagnostic fragment ion, and others are not. Additionally, some diagnostic fragment ions exhibit, in general, higher intensity than others. Nevertheless, the clustering analysis performed demonstrates that this approach can provide additional perception on the relationships of MS^2^ fragment ions and outline groups of parent ion → daughter ion.

## 3. Conclusions

Herein, utilizing the UHPLC-Orbitrap-MS technique, a total of 51 oleraceins were tentatively identified in hydromethanolic extracts of purslane; 25 of them are new structures. Diagnostic ion filtering (DIF) and diagnostic difference filtering (DDF) were employed to filter out MS data. Moreover, all 51 identified oleraceins were represented with a selected set of 43 diagnostic fragment ions that permitted the generation of a distance matrix. Furthermore, clustering of the identified oleraceins, based on their MS^2^ features, was achieved, and presented by heatmaps, pam and *k*-means clustering, principal component analysis, and hierarchical clustering. Here, we demonstrate that clustering methods can provide valuable insights in the MS^2^ elucidation of natural compounds in complex mixtures.

## 4. Materials and Methods

### 4.1. Plant Material

*Portulaca oleracea*, L. aerial parts were gathered from v. Orizovo, Bulgaria (42.208889° N–25.170278° E) and identified by one of us (V.B.). Voucher specimens were deposited at the Faculty of Pharmacy, Medical University of Sofia, Bulgaria (Herbarium Facultatis Pharmaceuticae Sophiensis № 1563–1574).

### 4.2. Extraction and Sample Preparation

The hydromethanolic extracts of purslane were obtained as described in our previous study [28] with small modifications. In brief, air-dried aerial parts of purslane were powdered, 3.00 g of plant material were extracted twice by sonication with 10 mL 80% MeOH at 50 °C for 15 min in an ultrasonic bath. The combined extracts were filtered and diluted to 25 mL in volumetric flasks with 80% MeOH. The solutions were filtered through a 0.22 μm syringe filter, and 1μL was injected into the LC instrument for LC-MS analysis.

### 4.3. UHPLC-HR-MS Instrument

The UHPLC system consisted of Dionex UltiMate 3000 RSLC HPLC, equipped with an SRD-3600 solvent rack degasser, an HPG-3400RS binary pump with solvent selection valve, a WPS-3000TRS thermostated autosampler, and a TCC-3000RS thermostated column compartment (Thermo Fisher Scientific, Germering, Germany). The UHPLC system was controlled by Chromeleon software, version 7.2. The effluents were connected on-line with a Q Exactive Plus mass spectrometer (Thermo Fisher Scientific, Bremen, Germany) equipped with a heated electrospray ionization (HESI-II) probe.

### 4.4. Chromatographic Parameters

Elution was carried out on Kromasil EternityXT C18 (1.8 μm, 2.1×100 mm, Akzo Nobel, Bohus, Sweden) column maintained at 40 °C. The chromatographic conditions were as described elsewhere [28] with slight modifications. The binary mobile phase consisted of A: 0.1% formic acid in water, and B: 0.1% formic acid in acetonitrile. The total run time was 23.5 min. The acquisition time where substances were analyzed with MS was 18 min and set between the 2nd and the 20th min. The following gradient was used: the mobile phase was held at 5% B for 0.5 min and then gradually turned to 33% B over 19.5 min. Next, % B was increased gradually to 95 % over 1 min and maintained at 95% B for 2 min. The system was turned to the initial condition of 5% B in 1 min and re-equilibrated over 4 min. Oleraceins eluted between the 5th and 17th min. The flow rate and the injection volume were set to 300 μL/min and 1 μL, respectively. 

### 4.5. Mass Spectrometric Parameters

For MS^2^ fragmentation analysis, several normalized collision energies (NCE) were tested to select the optimal conditions. The 20 NCE gave satisfactory abundance of variety of heavier fragment ions, and 40 NCE provided good intensity to lower *m*/*z* specific fragment ions, and thus, a stepped 20–40 NCE was selected for initial screening of oleraceins. Mass spectrometric parameters for Full-scan MS were as follows: resolution 17,500 (at *m*/*z* 200); AGC target 1.0×10^6^; Maximum IT 83ms; Scan range 500–2000 *m/z.* For dd-MS^2^, the following parameters were used: TopN 10; isolation window 1.0 *m*/*z*; stepped NCE 20–40; Minimum AGC target 8.0×10^3^; Intensity threshold 9.6×10^4^; Apex trigger 2 to 6 s; dynamic exclusion 3 s. The structural elucidation of the oleraceins was achieved by manual inspection of the MS^2^ spectra in Xcalibur 4.2 software (Thermo Fisher Scientific).

### 4.6. Mass Spectral Filtering by Diagnostic Ion Filtering (DIF) and Diagnostic Difference Filtering (DDF)

Initially, vendor *.raw (Thermo Fisher Scientific) files were converted to *.mzML files by msConvertGUI 3.0 (ProteoWizard) [31] and imported to MZmine 2.53 [32]. Then, DIF was applied based on the presence of two of the specific fragment ions for 5,6-dihydroxyindoline-2-carboxylic acid (called below in the text as “indoline core”): 194.0459 *m*/*z* (chemical formula: C_9_H_8_O_4_N^−^) and 150.0560 *m*/*z* (chemical formula: C_8_H_8_O_2_N^−^) (Figure 3), with a ±15 ppm threshold. MZmine also offers “diagnostic neutral loss” filtering for searching of specific mass difference(s) only between the precursor ion and each of its fragments. However, since we were interested in searching for the specific mass difference including between fragment ions of the same precursor ion (Figure 4), a DDF approach was applied to refine the selection of molecules that supposedly possess the 5,6-dihydroxyindoline-2-carboxylic acid substructure. DDF involved searching for a specific mass difference between each fragment (including the precursor ion, even if it was not present in the MS^2^ spectrum) and all lower *m*/*z* fragment ions. That difference was set to 195.05316 Da and suggested a neutral loss of 5,6-dihydroxyindoline-2-carboxylic acid. DDF was achieved by an in-house script written in Python 3.7.1 programming language. The defined threshold was set to ±15 ppm of the ions from which the difference originated. Thus, in the fragmentation transition 340.0848 *m*/*z* → 145.0300 *m*/*z*, with a threshold of ±15 ppm, the searched difference was between 145.0300 ±15 ppm and 340.0848 ±15 ppm (i.e., from 195.0475 Da to 195.0621 Da). Thus, if the difference originated from heavier fragments, a bigger mass threshold was used, and vice versa.

### 4.7. Grouping of MS^2^ Scans

In order to group MS^2^ scans that presumably derive from the same substance, MS^2^ scans with precursor ion *m*/*z* within 15 ppm and within 1.5 % deviation in retention times were added together, and afterwards manually checked. In these grouped MS^2^ scans, fragment ions that were within 15 ppm *m*/*z* were considered identical, their intensities added, and their masses recalculated by weighted mean averaging:
$${\left(m/z\right)}_{avg}=\frac{\sum_{i=1}^{N}int_{i}\times {\left(m/z\right)}_{i}}{N}\;$$
where ${\left(m/z\right)}_{avg}$ is the recalculated *m*/*z* value, ${\left(m/z\right)}_{i}$ and $int_{i}$ are the *m*/*z* and the intensity of the ith fragment ion, respectively. Fragment ions having less than 0.5% intensity and mass < 100 Da were excluded. The retention time of the precursor ion with the highest intensity was chosen as the retention time of grouped MS^2^ scans, i.e., the peak apex.

### 4.8. Clustering Methods

Clustering methods including heatmaps, *k*-means and pam clustering, principal component analysis (PCA) and hierarchical clustering were conducted in R 4.1.0 programming language [33] operated under RStudio (Version 1.4.1717, RStudio, PBC). Clustering visualizations were achieved using the package “factoextra” 1.0.7 [34]. Other used R packages include: “tidyverse” 1.3.1 [35], including packages “dplyr” 1.0.7 and “readr” 2.0.0; package “cluster” 2.1.2 [36]; package “igraph” 1.2.6 [37]; package “dendextend” 1.15.1 [38]. Distance matrix and hierarchical clustering were calculated with method = “euclidean” and method = “average”, respectively.

### 4.9. Used Abbreviations

For simplicity and clarity of the presentation, the following abbreviations are used throughout this paper: hydroxycinnamic acid: HCA; hydroxybenzoyl: hb or O; coumaroyl: coum or C; caffeoyl: caff or A; glucosyl: glu or G; feruloyl: fer or F; indoline core: ind or I; sinapoyl: sin or S. In case multiple oleraceins bore the same structure, the names of the compounds were suffixed with numbers, i.e., FGGIC, FGGIC.1, FGGIC.2, etc. (Table 2 and Table 3).

## Figures and Tables

**Figure 1 plants-10-01921-f001:**
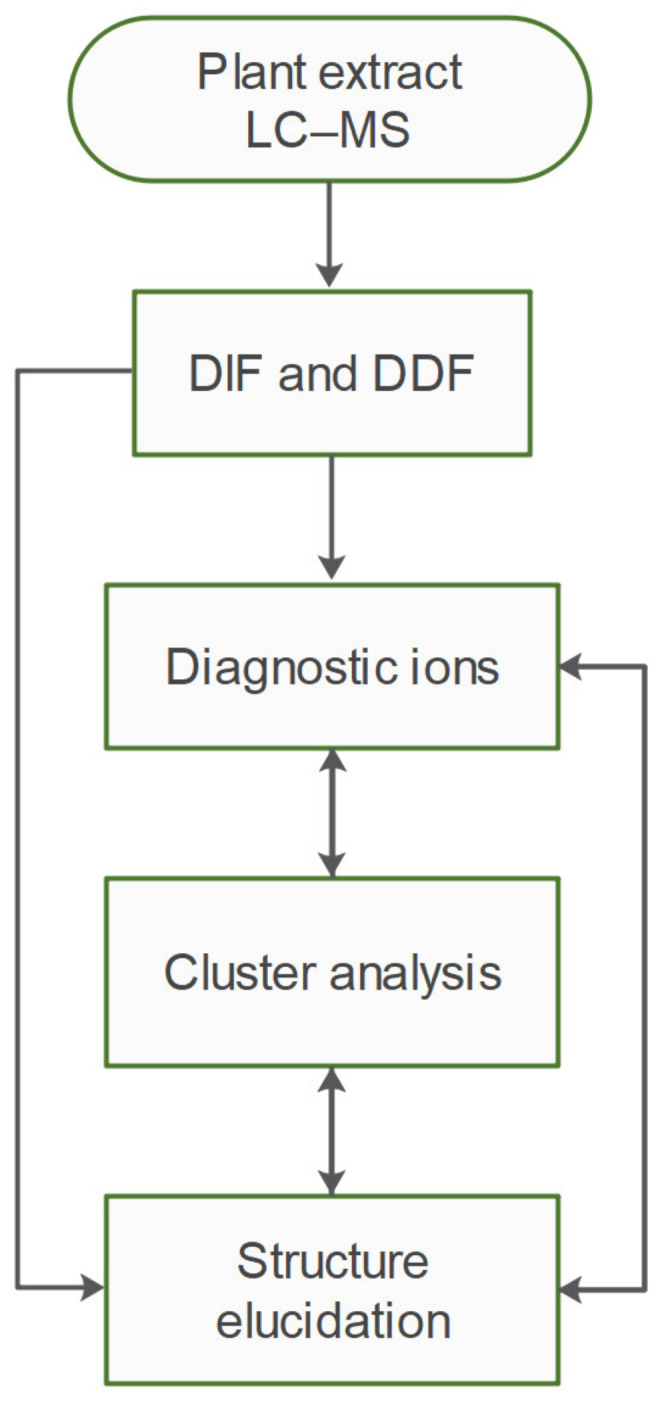
Workflow diagram.

**Figure 2 plants-10-01921-f002:**
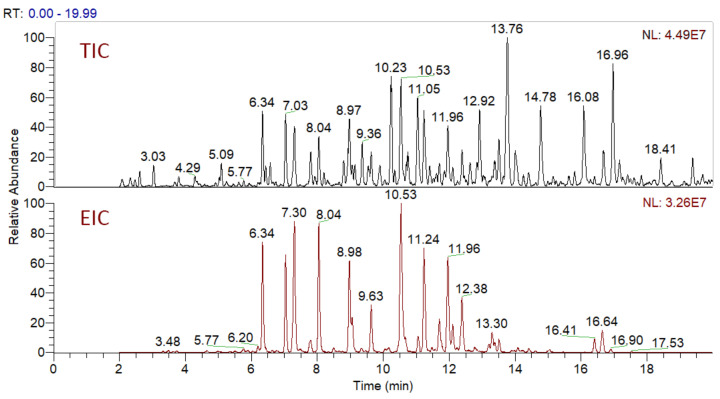
Total ion chromatogram (TIC) and extracted ion chromatogram (EIC) of the identified oleraceins. Oleraceins represent a major component of the hydromethanolic extract from purslane with very close normalization levels (NL) between TIC and EIC. Oleraceins eluted between the 5th and the 17th min.

**Figure 3 plants-10-01921-f003:**
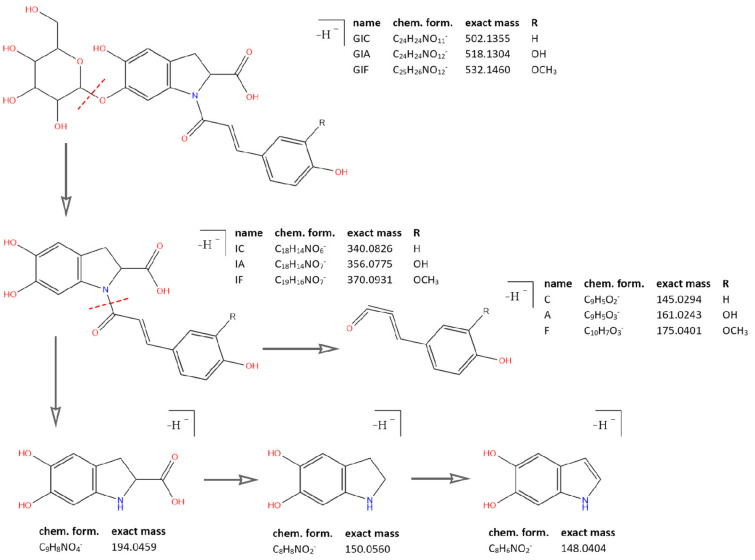
Common fragment ions of the three common substructures in oleraceins: GIC, GIA, and GIF.

**Figure 4 plants-10-01921-f004:**
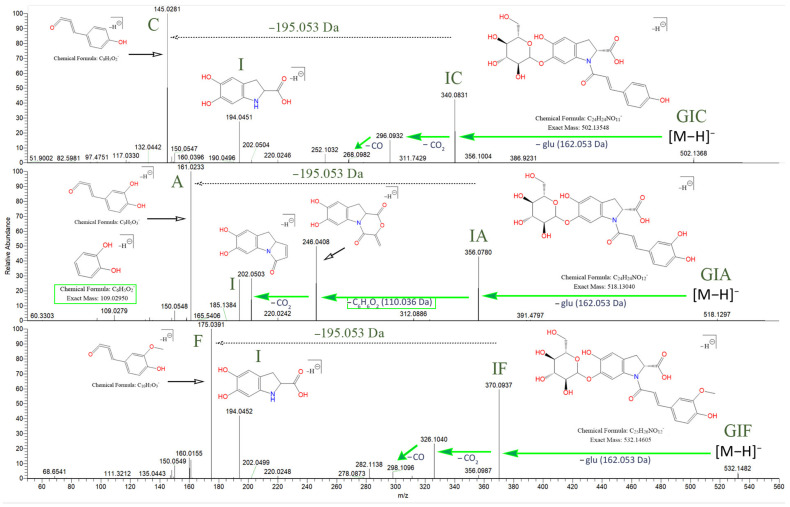
Fragmentation behavior for GIC, GIA, and GIF with proposed chemical structures. The searched specific difference of 195.05316 Da is marked, suggesting a neutral loss of 5,6-dihydroxyindoline-2-carboxylic acid (the indoline core). As shown, IA demonstrates a different fragmentation behavior, compared to IC and IF.

**Figure 5 plants-10-01921-f005:**
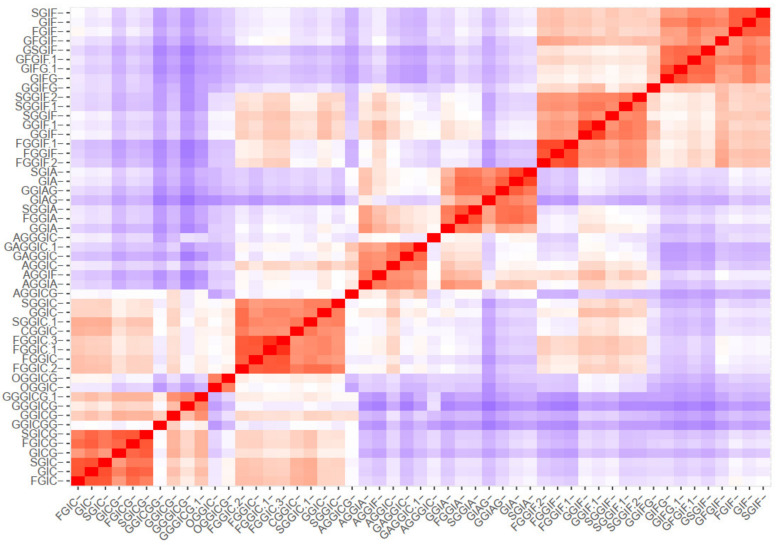
Ordered and unordered heatmaps based on a distance matrix using the MS^2^ data of the 51 identified oleracein (red—higher and blue—lower value).

**Figure 6 plants-10-01921-f006:**
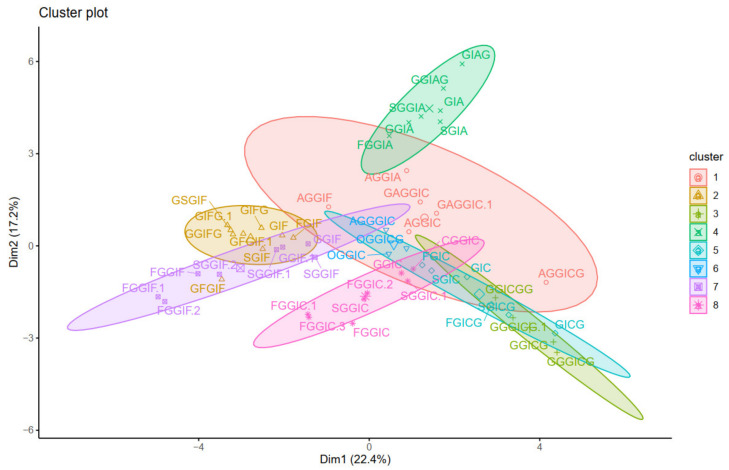
Visualization of the *k*-means clustering.

**Figure 7 plants-10-01921-f007:**
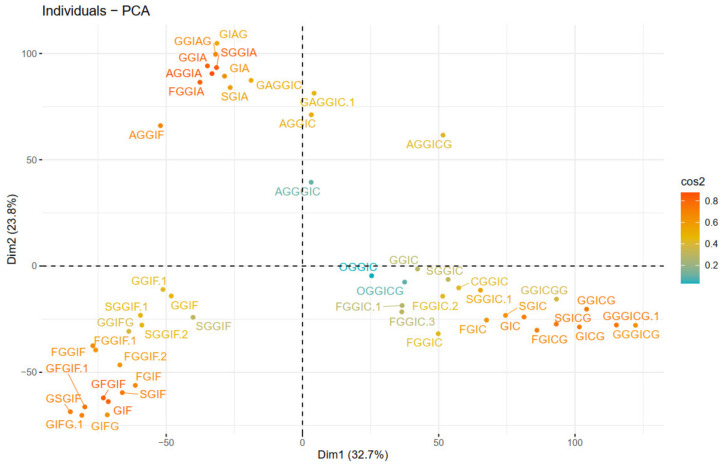
PCA plot of individuals (oleraceins).

**Figure 8 plants-10-01921-f008:**
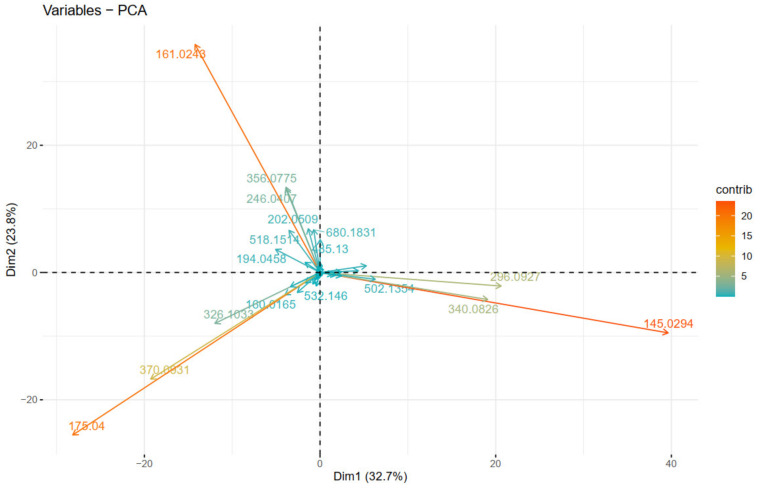
PCA plot of variables (fragment ions).

**Figure 9 plants-10-01921-f009:**
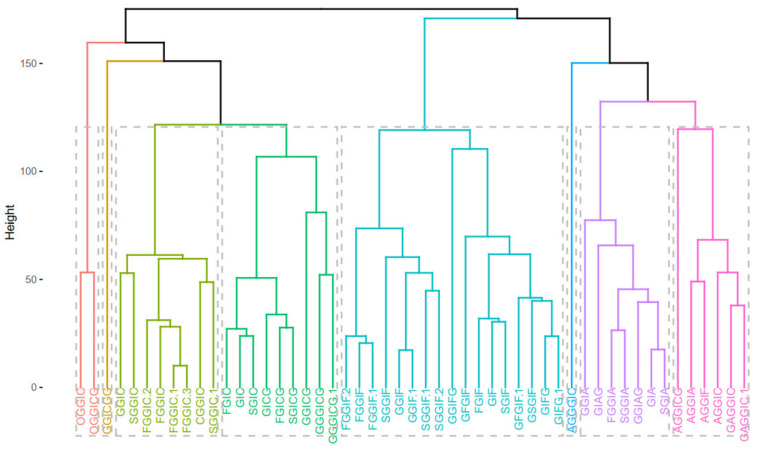
Dendrogram (**upper**) and phylogenetic tree-like dendrogram (**lower**) of the 51 identified oleraceins.

**Table 1 plants-10-01921-t001:** Known isolated and characterized oleraceins in the literature: A–D [29]; oleraceins E, T, U, V, W—not included; F and G [30]; H, I, K, L, N–S [27]; X–ZB [12].

a(Oleraceins A–D, F, G)					
**Name**	**R_1_**	**R_2_**	**R_3_**		** 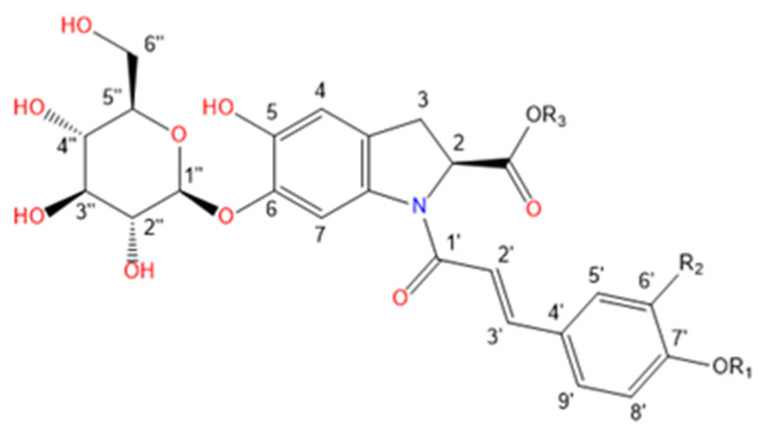 **
Oleracein A	H	H	H		
Oleracein B	H	OCH_3_	H		
Oleracein C	glu	H	H		
Oleracein D	glu	OCH_3_	H		
Oleracein F	H	OCH_3_	OCH_3_		
Oleracein G	H	H	OCH_3_		
b(Oleraceins H, I, K, L, N–S, X)					
**Name**	**R_1_**	**R_2_**	**R_3_**	**R_4_**	** 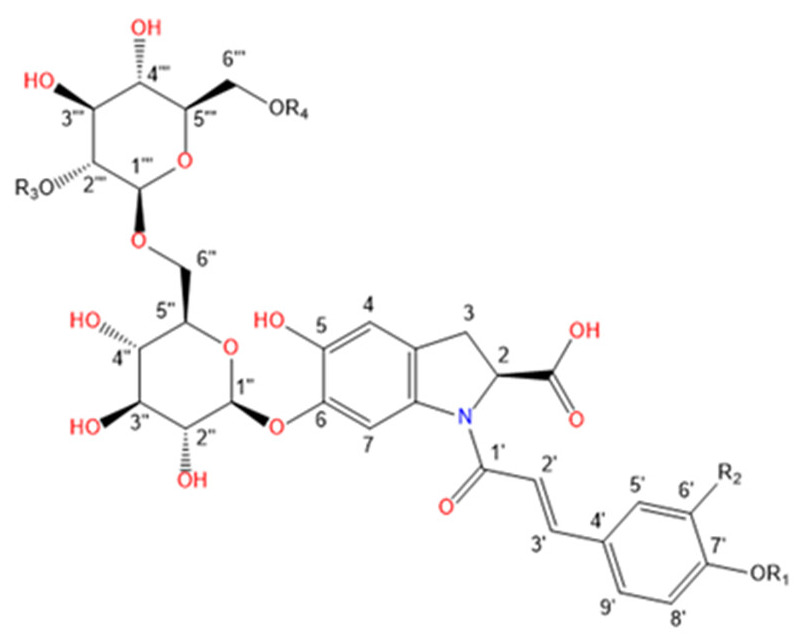 **
Oleracein H	H	H	H	H	
Oleracein I	H	OCH_3_	H	H	
Oleracein K	H	H	caff	H	
Oleracein L	H	OCH_3_	caff	H	
Oleracein N	H	H	fer	H	
Oleracein O	H	OCH_3_	fer	H	
Oleracein P	glu	H	H	H	
Oleracein Q	glu	OCH_3_	H	H	
Oleracein R	glu	H	fer	H	
Oleracein S	H	H	H	fer	
Oleracein X	H	OCH_3_	sin	H	
c(Oleraceins Y–ZB)					
**Name**	**R_1_**	**R_2_**	**R_3_**	**R_4_**	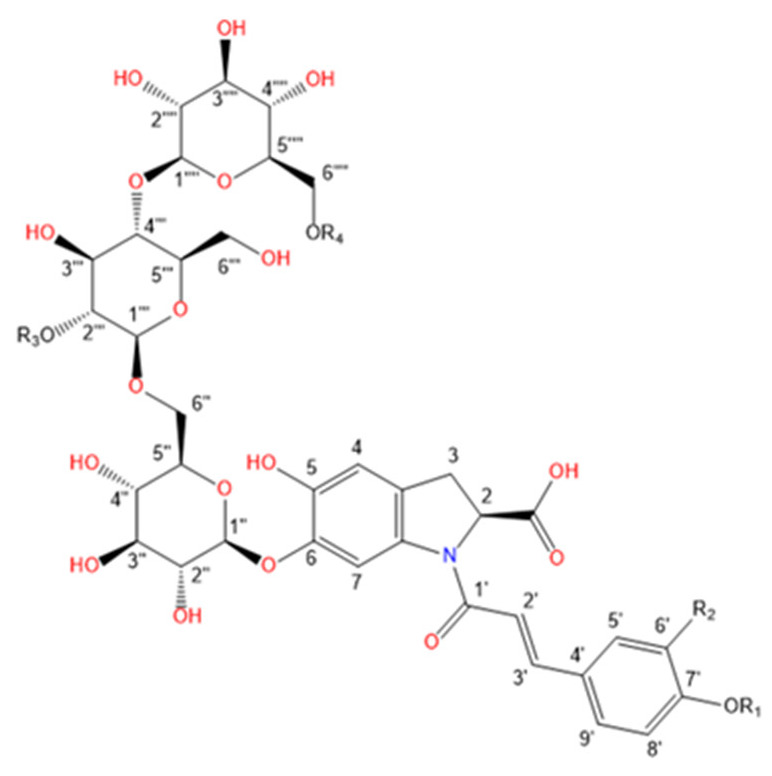
Oleracein Y	H	H	caff	caff	
Oleracein Z	H	H	caff	fer	
Oleracein ZA	H	OCH_3_	fer	H	
Oleracein ZB	H	OCH_3_	fer	sin	

**Table 2 plants-10-01921-t002:** Structure representation of the identified 51 oleraceins. For a list of used abbreviations, see “4.9. Used abbreviations”.

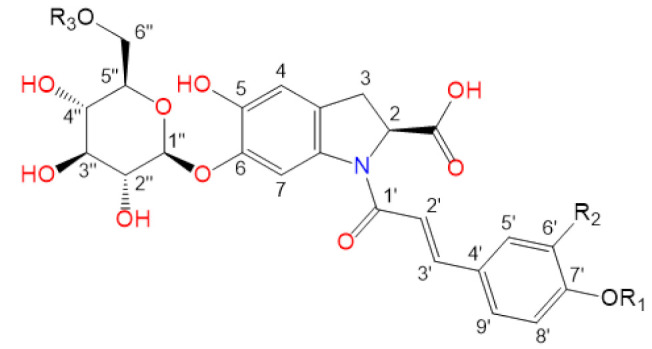						
	Compound	Abbrev	R_1_	R_2_	R_3_	Structure Match
**1**	glu-ind-coum	GIC	H	H	H	Oleracein A [29]
**2**	glu-ind-caff	GIA	H	OH	H	Oleracein W glu [28] *
**3**	glu-ind-fer	GIF	H	OCH_3_	H	Oleracein B [29]
**4**	glu-glu-ind-coum	GGIC	H	H	glu	Oleracein H [27]
**5**	glu-ind-coum-glu	GICG	glu	H	H	Oleracein C [29]
**6**	fer-glu-ind-coum	FGIC	H	H	fer	this paper
**7**	glu-ind-caff-glu	GIAG	glu	OH	H	this paper
**8**	glu-glu-ind-caff	GGIA	H	OH	glu	this paper
**9**	glu-ind-fer-glu	GIFG	glu	OCH_3_	H	Oleracein D [29]
**10**	glu-ind-fer-glu	GIFG.1	glu	OCH_3_	H	Oleracein D [29]
**11**	glu-glu-ind-fer	GGIF	H	OCH_3_	glu	Oleracein I [27]
**12**	glu-glu-ind-fer	GGIF.1	H	OCH_3_	glu	Oleracein I [27]
**13**	sin-glu-ind-coum	SGIC	H	H	sin	this paper
**14**	fer-glu-ind-fer	FGIF	H	OCH_3_	fer	this paper
**15**	sin-glu-ind-caff	SGIA	H	OH	sin	this paper
**16**	sin-glu-ind-fer	SGIF	H	OCH_3_	sin	this paper
**17**	hb-glu-glu-ind-coum	OGGIC	H	H	hb-glu	this paper
**18**	coum-glu-glu-ind-coum	CGGIC	H	H	coum-glu	this paper
**19**	caff-glu-glu-ind-coum	AGGIC	H	H	caff-glu	Oleracein K [27]
**20**	glu-glu-ind-coum-glu	GGICG	glu	H	glu	Oleracein P [27]
**21**	fer-glu-ind-coum-glu	FGICG	glu	H	fer	this paper
**22**	fer-glu-glu-ind-coum	FGGIC	H	H	fer-glu	Oleracein N/S [27]
**23**	fer-glu-glu-ind-coum	FGGIC.1	H	H	fer-glu	Oleracein N/S [27]
**24**	fer-glu-glu-ind-coum	FGGIC.2	H	H	fer-glu	Oleracein N/S [27]
**25**	fer-glu-glu-ind-coum	FGGIC.3	H	H	fer-glu	Oleracein N/S [27]
**26**	caff-glu-glu-ind-caff	AGGIA	H	OH	caff-glu	this paper
**27**	glu-glu-ind-caff-glu	GGIAG	glu	OH	glu	this paper
**28**	caff-glu-glu-ind-fer	AGGIF	H	OCH_3_	caff-glu	Oleracein L [27]
**29**	fer-glu-glu-ind-caff	FGGIA	H	OH	fer-glu	Oleracein J [26] *
**30**	glu-glu-ind-fer-glu	GGIFG	glu	OCH_3_	glu	Oleracein Q [27]
**31**	fer-glu-glu-ind-fer	FGGIF	H	OCH_3_	fer-glu	Oleracein O [27]
**32**	fer-glu-glu-ind-fer	FGGIF.1	H	OCH_3_	fer-glu	Oleracein O [27]
**33**	fer-glu-glu-ind-fer	FGGIF.2	H	OCH_3_	fer-glu	Oleracein O [27]
**34**	sin-glu-glu-ind-coum	SGGIC	H	H	sin-glu	Oleracein M [26] *
**35**	sin-glu-glu-ind-coum	SGGIC.1	H	H	sin-glu	Oleracein M [26] *
**36**	glu-fer-ind-fer-glu	GFGIF	H	OCH_3_	glu-fer	this paper
**37**	glu-fer-ind-fer-glu	GFGIF.1	H	OCH_3_	glu-fer	this paper
**39**	sin-glu-ind-coum-glu	SGICG	glu	H	sin	this paper
**40**	sin-glu-glu-ind-caff	SGGIA	H	OH	sin-glu	this paper
**38**	sin-glu-glu-ind-fer	SGGIF	H	OCH_3_	sin-glu	Oleracein X [12]
**41**	sin-glu-glu-ind-fer	SGGIF.1	H	OCH_3_	sin-glu	Oleracein X [12]
**42**	sin-glu-glu-ind-fer	SGGIF.2	H	OCH_3_	sin-glu	Oleracein X [12]
**43**	glu-sin-glu-ind-fer	GSGIF	H	OCH_3_	glu-sin	this paper
**44**	hb-glu-glu-ind-coum-glu	OGGICG	glu	H	hb-glu	this paper
**45**	glu-caff-glu-glu-ind-coum	GAGGIC	H	H	glu-caff-glu	this paper
**46**	glu-caff-glu-glu-ind-coum	GAGGIC.1	H	H	glu-caff-glu	this paper
**47**	caff-glu-glu-glu-ind-coum	AGGGIC	H	H	caff-glu-glu	this paper
**48**	caff-glu-glu-ind-coum-glu	AGGICG	glu	H	caff-glu	this paper
**49**	glu-glu-glu-ind-coum-glu	GGGICG	glu	H	glu-glu	this paper
**50**	glu-glu-ind-coum-glu-glu	GGGICG.1	glu-glu	H	glu	this paper
**51**	glu-glu-glu-ind-coum-glu	GGICGG	glu	H	glu-glu	this paper

* These references provide identification based on MS^2^ data only, and not NMR.

**Table 3 plants-10-01921-t003:** Chromatographic and mass spectral features of the identified 51 oleraceins.

	Abbrev	exact *m*/*z*	MS^2^	rt (min)	Structure Match
**1**	GIC	502.1355	502.1354 (3.57), 340.0826 (45.06), 296.0927 (16.56), 268.0978 (2.50), 202.0509 (1.70), 194.0458 (28.87), 150.0560 (5.99), 148.0403 (4.20), 145.0294 (100)	10.54	Oleracein A [29]
**2**	GIA	518.1304	518.1303 (0.75), 356.0775 (43.12), 246.0407 (49.23), 202.0509 (27.05), 194.0458 (32.49), 161.0243 (100), 150.0560 (8.43), 148.0403 (2.46)	9.07	Oleracein W glu [28]
**3**	GIF	532.1460	532.1460 (3.76), 370.0931 (60.33), 326.1033 (22.89), 298.1084 (4.41), 194.0458 (42.41), 175.0400 (100), 161.0243 (11.88), 160.0165 (13.6), 150.0560 (8.42), 148.0403 (5.95)	11.23	Oleracein B [29]
**4**	GGIC	664.1883	664.1882 (5.98), 518.1514 (65.37), 340.0826 (17.09), 296.0927 (35.29), 268.0978 (0.84), 202.0509 (2.79), 194.0458 (100), 150.0560 (34.78), 148.0403 (9.65), 145.0294 (56.75)	8.98	Oleracein H [27]
**5**	GICG	664.1883	502.1354 (36.75), 340.0826 (85.46), 296.0927 (29.99), 268.0978 (10.68), 202.0509 (2.70), 194.0458 (24.46), 150.0560 (2.80), 148.0403 (5.44), 145.0294 (100)	7.30	Oleracein C [29]
**6**	FGIC	678.1828	532.1460 (10.94), 502.1354 (0.61), 340.0826 (31.99), 337.0928 (11.78), 296.0927 (11.26), 202.0509 (1.50), 194.0458 (29.34), 193.0506 (3.38), 179.0349 (0.51), 175.0400 (12.68), 161.0243 (6.46), 160.0165 (2.07), 150.0560 (7.25), 148.0403 (5.18), 145.0294 (100)	16.63	this paper
**7**	GIAG	680.1832	518.1303 (25.28), 356.0775 (89.56), 246.0407 (66.10), 202.0509 (31.77), 194.0458 (19.02), 161.0243 (100), 150.0560 (2.52), 148.0403 (1.91)	7.26	this paper
**8**	GGIA	680.1832	680.1831 (1.22), 518.1514 (41.76), 356.0775 (45.60), 246.0407 (52.48), 202.0509 (28.04), 194.0458 (100), 161.0243 (89.61), 150.0560 (29.03), 148.0403 (7.30)	7.81	this paper
**9**	GIFG	694.1989	532.1460 (17.77), 370.0931 (100), 326.1033 (23.59), 298.1084 (16.57), 194.0458 (17.63), 175.0400 (78.68), 161.0243 (9.61), 160.0165 (3.28), 150.0560 (3.46), 148.0403 (9.41)	5.91	Oleracein D [29]
**10**	GIFG.1	694.1989	532.1460 (28.29), 370.0931 (100), 326.1033 (37.46), 298.1084 (19.50), 194.0458 (27.75), 175.0400 (86.22), 161.0243 (17.92), 160.0165 (6.93), 150.0560 (5.42), 148.0403 (7.20)	8.05	Oleracein D [29]
**11**	GGIF	694.1989	518.1514 (52.21), 370.0931 (24.35), 326.1033 (39.63), 194.0458 (100), 175.0400 (45.92), 161.0243 (18.24), 160.0165 (1.96), 150.0560 (27.72), 148.0403 (12.13)	6.78	Oleracein I [27]
**12**	GGIF.1	694.1989	694.1988 (9.12), 518.1514 (61.6), 370.0931 (24.57), 326.1033 (48.09), 298.1084 (0.65), 202.0509 (1.54), 194.0458 (100), 175.0400 (43.80), 161.0243 (22.26), 160.0165 (6.29), 150.0560 (31.35), 148.0403 (12.94)	9.64	Oleracein I [27]
**13**	SGIC	708.1934	562.1565 (6.41), 367.1034 (10.55), 340.0826 (30.31), 296.0927 (10.55), 268.0978 (0.71), 223.0611 (2.45), 205.0505 (9.60), 202.0509 (1.66), 194.0458 (26.35), 191.0349 (5.99), 150.0560 (8.36), 148.0403 (4.64), 145.0294 (100)	16.41	this paper
**14**	FGIF	708.1934	532.1460 (7.19), 370.0931 (35.83), 337.0928 (13.32), 326.1033 (14.97), 298.1084 (0.95), 194.0458 (30.33), 193.0506 (1.51), 175.0400 (100), 161.0243 (12.69), 160.0165 (17.46), 150.0560 (9.08), 148.0403 (6.21)	16.89	this paper
**15**	SGIA	724.1883	356.0775 (31.93), 246.0407 (42.83), 205.0505 (1.43), 202.0509 (25.69), 194.0458 (21.89), 161.0243 (100), 150.0560 (3.33), 148.0403 (1.96)	15.05	this paper
**16**	SGIF	738.2040	562.1565 (6.43), 532.1460 (0.56), 370.0931 (46.26), 367.1034 (14.50), 326.1033 (16.25), 298.1084 (1.23), 223.0611 (3.09), 205.0505 (15.1), 202.0509 (1.67), 194.0458 (38.53), 191.0349 (9.45), 175.0400 (100), 161.0243 (11.90), 160.0165 (20.90), 150.0560 (10.78), 148.0403 (8.52)	16.66	this paper
**17**	OGGIC	784.1883	638.1727 (79.93), 518.1514 (17.28), 443.1195 (23.46), 340.0826 (11.79), 194.0458 (98.68), 150.0560 (15.92), 148.0403 (11.58), 145.0294 (38.22), 137.0243 (92.77)	10.61	this paper
**18**	CGGIC	810.2251	664.1882 (30.64), 518.1514 (40.17), 518.1514 (19.88), 340.0826 (10.58), 296.0927 (5.81), 194.0458 (100), 150.0560 (24.01), 148.0403 (5.58), 145.0294 (92.62)	11.78	this paper
**19**	AGGIC	826.2200	680.1831 (27.17), 664.1882 (15.13), 518.1514 (78.8), 485.1300 (21.36), 356.0775 (1.33), 340.0826 (10.66), 296.0927 (22.75), 194.0458 (100), 179.0349 (8.31), 161.0243 (92.81), 150.0560 (26.74), 148.0403 (11.08), 145.0294 (36.92)	10.59	Oleracein K [27]
**20**	GGICG	826.2411	664.1882 (18.22), 518.1514 (36.79), 502.1354 (33.23), 340.0826 (59.20), 296.0927 (100), 268.0978 (10.38), 202.0509 (6.37), 194.0458 (37.51), 150.0560 (9.62), 148.0403 (9.67), 145.0294 (84.05)	6.34	Oleracein P [27]
**21**	FGICG	840.2357	532.1460 (8.85), 502.1354 (15.88), 340.0826 (62.02), 337.0928 (10.42), 296.0927 (26.3), 268.0978 (7.66), 202.0509 (3.70), 194.0458 (17.58), 193.0506 (2.93), 175.0400 (10.15), 161.0243 (3.16), 160.0165 (2.56), 150.0560 (3.66), 148.0403 (5.79), 145.0294 (100)	13.31	this paper
**22**	FGGIC	840.2357	694.1988 (14.74), 518.1514 (50.03), 502.1354 (2.97), 499.1457 (10.28), 340.0826 (31.03), 296.0927 (23.19), 268.0978 (2.50), 194.0458 (100), 193.0506 (21.57), 175.0400 (39.49), 150.0560 (25.63), 148.0403 (15.82), 145.0294 (96.89)	10.17	Oleracein N/S [27]
**23**	FGGIC.1	840.2357	694.1988 (23.48), 664.1882 (8.66), 518.1514 (53.69), 499.1457 (15.94), 485.1300 (1.22), 340.0826 (23.33), 296.0927 (23.23), 202.0509 (3.31), 194.0458 (100), 193.0506 (20.73), 175.0400 (32.55), 161.0243 (8.61), 160.0165 (7.44), 150.0560 (34.59), 148.0403 (13.45), 145.0294 (79.11)	11.95	Oleracein N/S [27]
**24**	FGGIC.2	840.2357	694.1988 (13.14), 664.1882 (4.40), 518.1514 (57.28), 499.1457 (4.07), 340.0826 (25.01), 296.0927 (23.14), 194.0458 (100), 193.0506 (13.25), 175.0400 (13.96), 161.0243 (2.36), 160.0165 (2.86), 150.0560 (27.76), 148.0403 (6.38), 145.0294 (85.71)	14.09	Oleracein N/S [27]
**25**	FGGIC.3	840.2357	694.1988 (22.64), 664.1882 (7.55), 518.1514 (58.14), 499.1457 (14.77), 485.1300 (1.82), 340.0826 (23.64), 296.0927 (25.22), 202.0509 (2.18), 194.0458 (100), 193.0506 (22.07), 175.0400 (34.66), 161.0243 (6.29), 160.0165 (7.44), 150.0560 (27.31), 148.0403 (15.84), 145.0294 (79.33)	12.36	Oleracein N/S [27]
**26**	AGGIA	842.2149	680.1831 (27.11), 518.1514 (65.15), 485.1300 (19.31), 356.0775 (23.66), 246.0407 (15.54), 202.0509 (4.30), 194.0458 (84.83), 179.0349 (3.45), 161.0243 (100), 150.0560 (18.25), 148.0403 (2.81)	9.34	this paper
**27**	GGIAG	842.2360	680.1831 (10.18), 356.0775 (67.36), 246.0407 (55.5), 202.0509 (32.63), 194.0458 (45.97), 161.0243 (100), 150.0560 (13.27), 148.0403 (6.58)	6.37	this paper
**28**	AGGIF	856.2306	694.1988 (16.68), 680.1831 (26.94), 518.1514 (70.4), 485.1300 (17.91), 370.0931 (11.80), 356.0775 (4.04), 326.1033 (25.35), 246.0407 (6.75), 202.0509 (3.29), 194.0458 (92.93), 179.0349 (8.50), 175.0400 (22.16), 161.0243 (100), 160.0165 (5.79), 150.0560 (32.09), 148.0403 (9.30)	11.09	Oleracein L [27]
**29**	FGGIA	856.2306	694.1988 (12.95), 518.1514 (23.06), 499.1457 (4.64), 356.0775 (40.26), 246.0407 (47.53), 202.0509 (26.56), 194.0458 (56.81), 193.0506 (7.4), 175.0400 (11.29), 161.0243 (100), 160.0165 (1.18), 150.0560 (17.86), 148.0403 (6.52)	10.69	Oleracein J [26]
**30**	GGIFG	856.2517	694.1988 (24.29), 532.1460 (16.47), 518.1514 (38.97), 370.0931 (46.21), 326.1033 (100), 298.1084 (9.02), 202.0509 (1.78), 194.0458 (35.00), 175.0400 (44.77), 161.0243 (28.71), 160.0165 (3.10), 150.0560 (6.95), 148.0403 (8.94)	7.04	Oleracein Q [27]
**31**	FGGIF	870.2462	694.1988 (31.87), 518.1514 (60.12), 499.1457 (5.37), 370.0931 (30.38), 326.1033 (29.67), 194.0458 (97.54), 193.0506 (21.72), 175.0400 (100), 161.0243 (25.47), 160.0165 (22.69), 150.0560 (24.82), 148.0403 (5.17)	10.65	Oleracein O [27]
**32**	FGGIF.1	870.2462	694.1988 (41.84), 518.1514 (63.14), 499.1457 (15.57), 485.1300 (1.83), 370.0931 (32.46), 326.1033 (31.88), 298.1084 (0.69), 202.0509 (1.89), 194.0458 (100), 193.0506 (27.19), 175.0400 (96.79), 161.0243 (21.81), 160.0165 (20.66), 150.0560 (31.54), 148.0403 (14.20)	12.40	Oleracein O [27]
**33**	FGGIF.2	870.2462	694.1988 (35.51), 518.1514 (58.24), 499.1457 (16.25), 370.0931 (38.93), 340.0826 (3.32), 326.1033 (30.96), 194.0458 (100), 193.0506 (22.11), 175.0400 (98.98), 161.0243 (21.41), 160.0165 (23.47), 150.0560 (31.12), 148.0403 (15.62), 145.0294 (17.78)	12.79	Oleracein O [27]
**34**	SGGIC	870.2462	724.2093 (23.36), 664.1882 (11.65), 529.1562 (16.21), 518.1514 (64.51), 340.0826 (24.79), 296.0927 (24.15), 268.0978 (0.85), 223.0611 (18.33), 205.0505 (30.61), 202.0509 (3.02), 194.0458 (100), 191.0349 (7.94), 150.0560 (34.8), 148.0403 (14.57), 145.0294 (77.17)	11.71	Oleracein M [26]
**35**	SGGIC.1	870.2462	724.2093 (9.68), 724.2093 (7.56), 664.1882 (8.12), 518.1514 (60.95), 340.0826 (18.71), 296.0927 (14.97), 205.0505 (8.40), 194.0458 (80.79), 150.0560 (13.70), 148.0403 (14.90), 145.0294 (100)	13.98	Oleracein M [26]
**36**	GFGIF	870.2462	724.2093 (11.32), 708.1933 (14.05), 694.1988 (9.76), 532.1460 (9.89), 518.1514 (21.93), 502.1354 (1.95), 499.1457 (2.86), 370.0931 (63.44), 337.0928 (2.41), 326.1033 (41.38), 194.0458 (61.09), 193.0506 (6.12), 175.0400 (100), 161.0243 (21.21), 160.0165 (11.32), 150.0560 (13.40), 148.0403 (41.38), 145.0294 (13.40)	14.33	this paper
**37**	GFGIF.1	870.2462	708.1933 (15.74), 532.1460 (27.33), 370.0931 (78.17), 337.0928 (13.75), 326.1033 (30.50), 298.1084 (6.09), 202.0509 (0.97), 194.0458 (32.87), 193.0506 (4.70), 175.0400 (100), 161.0243 (22.55), 160.0165 (12.61), 150.0560 (5.25), 148.0403 (8.80)	13.52	this paper
**38**	SGICG	870.2462	708.1933 (10.86), 562.1565 (8.41), 502.1354 (20.86), 367.1034 (9.61), 340.0826 (67.95), 296.0927 (25.95), 268.0978 (8.19), 223.0611 (1.97), 205.0505 (6.55), 202.0509 (3.52), 194.0458 (24.53), 191.0349 (3.58), 150.0560 (5.25), 148.0403 (6.75), 145.0294 (100)	13.20	this paper
**39**	SGGIA	886.2411	724.2093 (13.2), 518.1514 (23.7), 356.0775 (39.43), 246.0407 (47.59), 223.0611 (5.59), 205.0505 (5.59), 202.0509 (34.3), 194.0458 (61.58), 161.0243 (100), 150.0560 (15.36), 148.0403 (5.91)	10.46	this paper
**40**	SGGIF	900.2568	724.2093 (13.98), 518.1514 (57.73), 370.0931 (8.6), 223.0611 (11.15), 205.0505 (12.6), 194.0458 (100), 191.0349 (8.23), 175.0400 (64.27), 150.0560 (14.17), 148.0403 (12.00)	10.56	Oleracein X [12]
**41**	SGGIF.1	900.2568	694.1988 (13.27), 518.1514 (70.61), 370.0931 (39.16), 326.1033 (21.30), 223.0611 (9.33), 205.0505 (17.65), 194.0458 (94.20), 175.0400 (58.85), 161.0243 (15.23), 150.0560 (42.13)	14.09	Oleracein X [12]
**42**	SGGIF.2	900.2568	724.2093 (23.6), 694.1988 (15.95), 529.1562 (16.23), 518.1514 (66.44), 370.0931 (31.89), 326.1033 (31.2), 223.0611 (18.25), 205.0505 (30.74), 202.0509 (2.01), 194.0458 (100), 191.0349 (10.59), 175.0400 (64.59), 161.0243 (13.07), 160.0165 (12.38), 150.0560 (29.99), 148.0403 (15.08)	12.12	Oleracein X [12]
**43**	GSGIF	900.2568	562.1565 (14.07), 532.1460 (17.71), 370.0931 (93.17), 367.1034 (19.66), 326.1033 (29.11), 298.1084 (11.67), 205.0505 (10.66), 202.0509 (2.17), 194.0458 (29.35), 191.0349 (8), 175.0400 (100), 161.0243 (25.28), 160.0165 (14), 150.0560 (2.55), 148.0403 (3.44)	13.41	this paper
**44**	OGGICG	946.2623	638.1727 (100), 518.1514 (21.76), 443.1195 (14.11), 340.0826 (9.03), 296.0927 (12.12), 194.0458 (72.48), 150.0560 (21.8), 145.0294 (50.83), 137.0243 (60.3)	8.65	this paper
**45**	GAGGIC	988.2728	842.2359 (9.79), 680.1831 (53.3), 664.1882 (14.61), 518.1514 (90.13), 485.1300 (44.78), 340.0826 (7.3), 296.0927 (8.35), 194.0458 (79.91), 179.0349 (21.21), 161.0243 (100), 150.0560 (29.14), 145.0294 (11.9)	9.45	this paper
**46**	GAGGIC.1	988.2728	842.2359 (13.29), 826.2199 (12.04), 680.1831 (64.32), 664.1882 (11.38), 518.1514 (85.83), 485.1300 (39.72), 340.0826 (10.79), 296.0927 (26.65), 194.0458 (77.31), 179.0349 (8.74), 161.0243 (100), 150.0560 (21.07), 148.0403 (4.19), 145.0294 (37.59)	8.52	this paper
**47**	AGGGIC	988.2728	842.2359 (34.71), 826.2410 (10.08), 680.2043 (100), 340.0826 (2.65), 296.0927 (5.14), 194.0458 (70.43), 161.0243 (62.8), 150.0560 (27.02), 148.0403 (13.54), 145.0294 (18.12)	10.07	this paper
**48**	AGGICG	988.2728	826.2410 (20.35), 680.1831 (19.77), 664.1882 (38.74), 518.1514 (49.86), 502.1354 (32.18), 485.1300 (44.02), 340.0826 (42.12), 296.0927 (71.42), 268.0978 (6.94), 202.0509 (4.21), 194.0458 (39.22), 179.0349 (11.46), 161.0243 (100), 150.0560 (9.28), 148.0403 (8.84), 145.0294 (56.04)	7.78	this paper
**49**	GGGICG	988.2940	988.2939 (23.62), 826.2410 (32.04), 680.2043 (76.98), 502.1354 (34.71), 340.0826 (63.9), 296.0927 (97.09), 268.0978 (12.36), 202.0509 (6.24), 194.0458 (42.82), 150.0560 (11.81), 148.0403 (16.66), 145.0294 (100)	6.21	this paper
**50**	GGGICG.1	988.2940	826.2410 (23.97), 680.2043 (37.17), 502.1354 (29.62), 340.0826 (51.79), 296.0927 (98.85), 194.0458 (35.22), 150.0560 (10.56), 148.0403 (7.48), 145.0294 (100)	6.03	this paper
**51**	GGICGG	988.2940	988.2939 (52.98), 664.1882 (98.49), 518.1514 (35.65), 340.0826 (18.49), 296.0927 (100), 194.0458 (26.65), 150.0560 (4.70), 148.0403 (5.22), 145.0294 (75.66)	5.83	this paper

## Data Availability

Not applicable.

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
