# Peer review of "UHPLC-Orbitrap-MS Tentative Identification of 51 Oleraceins (Cyclo-Dopa Amides) in Portulaca oleracea L. Cluster Analysis and MS2 Filtering by Mass Difference"

_plants, 2021, doi:10.3390/plants10091921_

Round 1

Reviewer 1 Report

This manuscript is quite interesting, that the authors tentatively identified 51 oleraceins from purslane using UHPLC-MS-MS analysis and classified these compounds according to chemometric analysis of their MS2 data! This work will inspire the phytochemists to further investigate these compounds and thoroughly elucidate their absolute structures. This work is also valuable for future analytical and pharmacological study. Some problems were listed below.

Page 1, line 42: Error! Reference source not found.. 42

Deleted. The reference “Journal of Chinese Pharmaceutical Sciences, 2014, 23(8):533-542” is provided.

Page 2, line 43: explanation for “oleraceins E, T, U, V, W are not included”.

As a beginning researcher in oleraceins, I should mentioned some history on this kind of alkaloids. The author also mentioned some of the history of these compounds in the manuscript.

The structure of oleracein E is a tetrahydroisoquinoline alkaloid, quite different from other oleracein compounds (indoline alkaloids). It was published for the first time in 2005 (Phytochemistry, 2005), together with olereacins A-D.

Following HPLC-MS-MS analysis led to the discovery of oleraceins F-G. The structure of oleraceins F-G were elucidated by NMR data in 2009.

In 2015, there was another paper reported the identification of oleraceins H-O from purslane using HPLC-MS-MS method (Journal of Chinese Pharmaceutical Sciences, 2014, 23(8):533-542). In 2012, oleraceins H, I, K, L, N-S (except oleraceins J and M) were isolated and their structures were thoroughly elucidated based on NMR and CD data (Journal of Natural Products, 2015). Since then, our group didn’t isolated and identified other oleraceins. In 2021, another group isolated and identified oleraceins X, Y, Z, ZA and ZB. And the authors of this manuscript also tentatively identified oleraceins U and W-glu (Line 54, page 3).

That is the reason why the isolation of oleraceins T, U, V, W were not included in the reference.

Page 2, line 43:

  • since R3 and R4 were noted different from those in the reference, some structures were written wrong in the second table.

Oleracein I was wrong, R3 is H.

Oleracein K was wrong, the position of H and caff should be exchanged.

Oleracein O was wrong, R1 is H, R2 is OCH3, R3 is fer.

Oleracein Q was wrong, R1 is glu, R3 is H.

  • since R1-R4 were noted different from those in the reference, the structure of Oleracein Y, Z, ZA, ZB were noted wrong in the third table, please check the original figures in the reference.

Page 2, line 48: deleted “, and E”

Page 3, line 92: Error! Reference source not found.

Please add the reference, Journal of Natural Products, 2015.

Page 3, line 94: “regarded as glucose” is more scientific than “regarded asβ-D-glucopyranoses”, since there were no NMR data to support it.

Page 3, line 97: “502.135 [M-H]- m/z” or “m/z 502.135 [M-H]- ? Please unify the position of m/z in all the manuscript.

Page 3, line 97: I cannot understand why “Other reported in the literature oleraceins with lower mass [29, 30] were not observed in our sample”.

Page 11, line 134: C ()?

Page 13, line 269: identified as?

Page 32, line 955: the first sentence is not complete.

Page 32, line 989: Please correct “hydroxybenzoyl: hydroxycinnamic acid: HCA ; hb or O; ” to “hydroxybenzoyl: hb or O; hydroxycinnamic acid: HCA; ” 

References:

  1. Some minor problems were highlighted yellow in the reviewed manuscript.
  2. reference 26 is “Journal of Chinese Pharmaceutical Sciences, 2014, 23(8):533-542”

Supplementary figure:

There are only some tables to reflect ESI-MS data in the supplementary material. Please give the original UHPLC spectrum and TIC spectrum of the purslane, also provide the ESI-MS and MS/MS spectrum for each compound. It would spend a lot of pages, but these original figures were quite important for your discovery and conclusion.

Reviewer 2 Report

This is a high quality manuscript presenting and describing the strategy for annotation of novel compounds, which has great value for plant metabolomics. The paper does not include any novel findings; rather, it represents a report of essential steps for improving reliability of novel identification. This manuscript should be published in this journal. My recommendations and suggestions are as following:

 - Main recommendation and suggestions:

Authors used HRMS and HRMS/MS data for structure elucidations of novel structures. Solely this method cannot provide “identification” or “Level 1 annotation”, so title should be changed accordingly. This method provides “Level 2 annotation” or “tentative identification” as it correctly mention in text, for ex. LINE#17 or #54. However, it also should be taken into account while discussing Table 2, since ref. #28 and #26 deal with MS data “only”, while ref. #27, #28 and #12 represent more detailed studies (including NMR and/or X-ray).

Minor comments:

  1. LINE 2: “Orbitrap-HRMS” is repetitive, either “Orbitrap” or “HRMS(/MS)”
  2. LINE #104: 1.5E04 is it total ion current or base peak?
  3. Table 1: I would use “this paper” instead “new”
  4. LINE #139 and bellow: HRMS data is reported with four digits after decimal point (like in table 1).
  5. Fig 2 (x1), Fig 3 (x3): Structure of “coumaroyl ion”, “C/A/F” cannot have triple bound and aldehyde. It should be O=C=C=C-o-Ph-OH or O=C=C=Cc1ccc(O)cc1 in SMILES.
  6. LINE #596, #597, #600: NCE is relative measure, i.e.
  7. LINE #598: The resolution of Orbitrap-based instruments should be defined as “17,500 (FWHM) at m/z 200” or “17,500 (at m/z 200).”
  8. Pages 24-34 repeated twice.
  9. Other small typos are in LINE #42, #92, #269, #409, #509

Reviewer 3 Report

The mauscript highlights that of the 51 identified structures, 25 are new structures, belonging to the class of oleraceins. This demonstrates how new compounds can be found from plants which are also useful for the prevention and treatment of various diseases. I suggest adding these manuscripts in the introduction to emphasize the importance of medicinal plants

Goni O, Khan MF, Rahman MM, Hasan MZ, Kader FB, Sazzad N, Sakib MA, Romano B, Haque MA, Capasso R. Pharmacological insights on the antidepressant, anxiolytic and aphrodisiac potentials of Aglaonema hookerianum Schott. J Ethnopharmacol. 2021 Mar 25;268:113664.

Khan MF, Kader FB, Arman M, Ahmed S, Lyzu C, Sakib SA, Tanzil SM, Zim AFMIU, Imran MAS, Venneri T, Romano B, Haque MA, Capasso R. Pharmacological insights and prediction of lead bioactive isolates of Dita bark through experimental and computer-aided mechanism. Biomed Pharmacother. 2020 Nov;131:110774.

Iqbal J, Abbasi BA, Ahmad R, Mahmoodi M, Munir A, Zahra SA, Shahbaz A, Shaukat M, Kanwal S, Uddin S, Mahmood T, Capasso R. Phytogenic Synthesis of Nickel Oxide Nanoparticles (NiO) Using Fresh Leaves Extract of <i>Rhamnus triquetra</i> (Wall.) and Investigation of Its Multiple In Vitro Biological Potentials. Biomedicines. 2020 May 12;8(5):117.

Are these compounds toxic? do the authors have any indications?

I suggest the authors to add the graphical abstract, so the results are more immediate

The manuscript would benefit from inclusion of introducing/bridging sentences between the individual parts of the "Results" that explain the logical order and rationale for the experiments
